# Signal detection in global mean temperatures after "Paris": an uncertainty and sensitivity analysis

Hans Visser[1], Sönke Dangendorf[2], Detlef P. van Vuuren[1,3], Bram Bregman[4] and
Arthur C. Petersen[5]

[1] PBL Netherlands Environmental Assessment Agency, Bilthoven, The Netherlands

[2] Research Institute for Water and Environment, University Siegen, Siegen, Germany

[3] Faculty of Geosciences, University Utrecht, Utrecht, The Netherlands

[4] Institute for Science, Innovation and Society, Radboud University, Nijmegen, The Netherlands

[5] STEaPP, University College London, London, Great Britain

*Correspondence to*: Hans Visser (hans.visser@pbl.nl)

**Abstract.** In December 2015, 195 countries agreed in Paris to 'hold the increase in global mean surface temperature (GMST) well below 2.0 °C above pre-industrial levels and to pursue efforts to limit the temperature
increase to 1.5 °C'. Since large financial flows will be needed to keep GMSTs below these targets, it is important to know how GMST has progressed since pre-industrial times. However, the Paris Agreement is not conclusive as regards methods to calculate it. Should trend progression be deduced from GCM simulations or from instrumental records by (statistical) trend methods? Which simulations or GMST datasets should be chosen, and which trend models? What is 'pre-industrial', and finally, are the Paris targets formulated for total warming, originating from
both natural and anthropogenic forcing, or do they refer to anthropogenic warming only? To find answers to these questions we performed an uncertainty and sensitivity analysis where datasets and model choices have been varied. For all cases we evaluated trend progression along with uncertainty information. To do so, we analysed four trend approaches and applied these to the five leading observational GMST products. We find GMST progression to be largely independent of various trend model approaches. However, GMST progression is significantly influenced
by the choice of GMST datasets. Uncertainties due to natural variability are largest in size. As a parallel path, we calculated GMST progression from an ensemble of 42 GCM simulations. Mean progression derived from GCM-based GMSTs appears to lie in the range of trend-dataset combinations. A difference between both approaches appears to be the width of uncertainty bands: GCM simulations show a much wider spread.  Finally, we discuss various choices for pre-industrial baselines and the role of warming definitions. Based on these findings we
propose an estimate for signal progression in GMSTs since pre-industrial.

## 1. Introduction

Global mean surface temperature (GMST) is undoubtedly one of the key indicators of climate change. Tollefson (2015) denotes the GMST indicator as 'the global thermostat'. Over the years many articles have been published in relation to GMST series and the patterns therein. These patterns combine an anthropogenic signal – induced by growing concentration of greenhouses and processes such as aerosol cooling – as well as natural variability. Natural variability can be regarded as a correlated noise process consisting of (i) internal random unforced (chaotic) variability and (ii) external radiatively forced changes. Here, internal variability is steered by short-term processes such as weather in the high latitudes or El Niño and La Niña, as well as by decadal processes such as the Interdecadal Pacific Oscillation (e.g., Trenberth, 2015; Fyfe et al. 2016; Xie, 2016; Meehl et al., 2016), and will result in correlated noise in GMSTs (Mudelsee, 2014; Roberts et al., 2015). Externally forced variability is mainly due to volcanic eruptions and variations in solar irradiance. It influences global temperatures on annual to centennial scales (IPCC, 2013 - Ch. 10; Forster et al., 2013; Mann et al., 2016). A recent realization of internal variability led to a fierce debate in the popular media: GMSTs were showing a claimed "slowdown", "pause" or "hiatus" from the year 1998 onwards (e.g., Lewandowski et al., 2015; Hedemann et al., 2017; Medhaug et al., 2017 - their figure 1).

GMST has been a crucial indicator in climate negotiations for a long time and it has even become more so at the following 21st Conference of Parties (COP21) in Paris, December 2015. The final accord, approved by 195 countries, agreed on GMST targets which aim to avoid increases of 1.5 and 2.0 °C compared to pre-industrial temperatures (UN, 2015). IPCC (2014a) showed that meeting such GMST targets will require deep reductions of GHG emissions at the cost of high investments in mitigation measures worldwide. Given the fact that all goals are formulated on the basis of this single GMST indicator, the question arises: what is the current GMST level since pre-industrial?

So far, little attention has been paid to this topic. IPCC (2013), in its attempt to clarify the meaning of GMST measurements, applied linear trends to three different GMST datasets. They reported a trend progression Δμ of 0.85 [0.65, 1.06] °C for the period 1880-2012. The uncertainty range stands for 90% confidence limits, originating from differences in datasets, natural variability of the climate system (forced and unforced). Hawkins et al. (2017) and Schurer et al. (2017) addressed the topic of trend progression *since pre-industrial* and quantified the role of various choices for pre-industrial baselines.

Hawkins et al. found that the period 1720-1800 would be the most suitable in physical terms, despite incomplete information about radiative forcings and very few direct observations during this time. Additionally, they concluded that the 1850-1900 period would be a reasonable surrogate for pre-industrial GMSTs, being only 0.05 °C warmer than the 1720-1800 period. Subsequently, Hawkins et al. analyzed GMST progression since pre-industrial by calculating the GMST mean over the 20-year period 1986-2005 for various GMST products and other instrumental data (their figure 4). Trend progression itself was approximated in the study by multiple regression

models with non-stationary explanatory variables such as historic GHG forcing curves or local temperature series (the Central England Temperature series or the De Bilt series). Schurer et al. found that GHGs had a significant warming effect on global temperatures if the period 1401-1800 is compared to 1850-1900: from 0.02 to 0.20 ºC (90% confidence limits). If all forcings are combined (GHG, solar, volcanic), they found a similar warming effect of 0.09 [0.03 - 0.19] °C.

In this article, we build on the work of Hawkins et al. but we do not base our GMST progression estimates on linear regression models with non-stationary regressors. The drawback of this approach is simply the linearity assumed, while the climate system is (highly) non-linear with a number of feedback processes. The same holds for the approach proposed by Otto et al. (2015) and Haustein et al. (2017) who apply temperature responses to (i) human-induces forcings and (ii) natural drivers as explanatory variables in a multiple regression model where the dependent variable is given by one of the observational GMST datasets.

Therefore, we follow two other trend estimation approaches: (i) statistical trend models and (ii) global temperature trends derived from Global Climate Models (GCMs). Furthermore, we avoid methods or presentations based on subjectively selected time-windows (such as Moving Averages). The drawback of time windows is that averages over 21-year periods or similar do not give estimates for the beginning and ending of the sample period chosen (thus, we would have no trend estimates for the period 2007-2016).

A final topic we address is that of warming definitions. Should the Paris targets be interpreted as warming due to both anthropogenic and natural forcings, or as warming due to anthropogenic warming only? The terms 'global warming' or 'total warming' are interpreted in most literature as the sum of anthropogenic warming plus long-term (decadal to centennial) natural warming, consistent with the IPCC definition of climate change (IPCC Annex II, 2014). However, some researchers interpret 'global warming' as anthropogenic warming only, consistent with the definition proposed by UNFCCC in their article 1 (Otto et al., 2015; Haustein et al. 2017; Millar et al. 2017). In both definitions, short-term natural variability – such as seen in "the hiatus period" – is smoothed from warming trends.

Our approach is that of an uncertainty and sensitivity analysis as promoted by Saltelli et al. (2004), Saisana et al. (2005) and Visser et al. (2015). We ask the following four major questions:

- How robust are estimates for GMST progression to specific choices of trend modelling, use of GCMs and specific choices of GMST datasets?
- How do these choices influence uncertainties in GMST progression in relation to uncertainties due to forced and unforced natural variability?
- Does the choice for a specific pre-industrial baseline or period play a role?
- Does it matter if we interpret the Paris targets as total warming or as anthropogenic warming only?

Since there is no 'true' or 'best' trend approach (Visser et al., 2015), we explore four trend methods and apply these to five leading GMST products (similar to Hawkins et al.). This leads to a 4-by-5 matrix of GMST trend progressions since 1880. As a parallel path, we compare these trend progressions to those deduced from GCMs. We analyse an ensemble of 42 GCM experiments from the Coupled Model Intercomparison Project phase 5 (CMIP5). GCMs are for a large part physics-based, in contrast to trend methods. However, there are also drawbacks, the main one being that GCMs are only approximations to the real climate system and have considerable biases. Although GCMs are tuned to meet the main characteristics of the present climate (Voosen, 2016), GMSTs derived from GCMs still exhibit a wide range of trend-progression estimates, as we will show.

In the discussion section, we address the role of various assumptions as for pre-industrial baselines, and differences in trend progression if Paris targets are interpreted as 'total warming' versus 'anthropogenic warming'.

Our analysis is confined to historic data only (up to and including 2016). Examples for GMST projections have been given by IPCC (2013 - Ch. 12), Forster et al. (2013), Mann (2014) and Schurer et al. (2017). A short-term prediction model is given by Suckling et al. (2016). An example of an uncertainty and sensitivity analysis of GMST projections has been given by Visser et al. (2000).

## 2 Data and methods

### 2.1 Data

Various research groups have published global GMST datasets. IPCC (2013 - section 2.4.3) used three datasets, namely the HadCRUT4 series (Morice et al., 2012; Hope, 2016), the NOAA dataset (Vose et al., 2012) and the NASA/GISS dataset (Hansen et al., 2010). In the analysis here, we instead use a recent update of the NOAA data (Karl et al., 2015). Karl et al. applied a number of corrections which mainly deal with sea surface temperatures, such as the change from buckets to engine intake thermometers. In addition, we added two series, i.e. the version of the HadCRUT4 data in which the missing data have been filled in as published by Cowtan and Way (2014) and the GMST series by Rohde et al. (2013). Note that these datasets are not independent. They start from roughly the same station data over land, and more importantly are based on only two SST analyses: HadSST3 and ERSSTv4.

Cowtan and Way re-analysed the HadCRUT4 series by applying a statistical interpolation technique (Kriging) and satellite data for regions where data are sparse. Their series shows higher GMST values in recent decades than the non-interpolated HadCRUT4 series due to the more-than-average warming of the poles. The land part of the GMST data of Rohde et al. (2013; Berkeley Earth group of researchers) systematically addressed major concerns of global warming sceptics, mainly dealing with potential bias from data selection, data adjustment, poor station quality and the urban heat island effect. The ocean part (about 70%) is taken from HadSST3. A summary of observational data products is given in table 1.

Since two out of five GMST products start in the year 1880, we use the period 1880-2016 as our period of analysis. We return to this point in the discussion section. All data were downloaded from the institution websites with 2016 as the final year.

Next to these instrumental-data based GMSTs we analyze three sets of GCM simulations all taken from CMIP5 (Taylor et al., 2012; IPCC, 2013 – Ch. 9-12). GMST is defined here as the global average of near-surface temperature (temperature at surface or 'tas' in short), in contrast to the observational datasets that use SST over sea for practical reasons (also denoted as 'blended temperature series'; Cowtan et al., 2015). The first set consists of GCM simulations where the input of greenhouse gases from 2005 onwards is taken from three Representative Concentration Pathways (RCPs): 4.5, 6.0 and 8.5 W/m$^2$ (Van Vuuren et al., 2011; IPCC, 2014 - section 12.4 and figure 12.5). These simulations cover the period 1861-2100. We have taken a set of 42 GCM simulations with one member per model for emission scenario RCP4.5 (simulations for the other RCPs partly overlap with this set and are not considered here). GMSTs from CMIP5 simulations are based on wide range of modeling differences such as climate sensitivities, cloud parametrization and aerosol forcing (e.g., IPCC 2013 - Ch. 9).

The second set that we have analyzed, consists of 37 GCM runs for natural variability, denoted as 'historicalNat'. These runs comprise forced and unforced natural variability but no GHG forcing (1860-2005). See Forster et al. (2013) for details. Finally, we analyzed 41 Pre-industrial Control (PiControl) runs with lengths varying between 200 and 1000 years. These runs simulate natural internal variability only. All CMIP5 runs were downloaded from the KNMI Climate Explorer website with one member per model (Trouet and Van Oldenborgh, 2013).

## 2.2 Trend modeling

The tracking of signals or trends in GMST series has a long history. Callender (1938) studied in detail zonal and global temperatures, along with estimates for warming due to greenhouse gases (figure 1). To smooth changes he used moving averages with a window of 10 years. A wide range of methods have been applied since then to isolate long-term signals or 'trends' in GMSTs. We have summarized trend techniques in Appendix A (table A.1).

As stated in the Introduction we choose statistical trend methods that allow for the quantification of trend progression where no window is needed and where uncertainty estimates are available for any incremental trend value. Furthermore, no specific period for pre-industrial has to be chosen (such as the mean of the 1851-1900 period or similar). 'Pre-industrial' is reflected in the choice of the start of the sample period only.

Based on these considerations we have selected four trend approaches for our sensitivity analysis: Ordinary Least Squares (OLS) linear trends, Integrated Random Walk (IRW) trends and two approaches with splines. The first trend - a linear fit by OLS - was chosen by IPCC (2013) as their main method. Uncertainties simply follow from the linear model:

$$var(\Delta\mu_{2016}) = var([a+b*2016] - [a+b*1880]) = var(137*b) = 137^2 * var(b) \qquad (1)$$

where 'a' is the intercept and 'b' the slope. The variance of 'b' follows from the OLS equations. Next to that the variance estimate is corrected by calculating effective sample sizes $N_{eff}$, based on annual data. This correction is important since residuals are not white noise due to persistence in natural processes. The signal is therefore considered as noise with a large decorrelation scale in this approach. The $N_{eff}$ correction method has been explained by Zieba (2010), Chandler and Scott (2011, Section 3.3.3) and IPCC (2013 - 2SM).

The second trend approach that fulfils our uncertainty requirements, are sub-models from the class of Structural Time Series models (STMs), in combination with the Kalman filter (Harvey, 1989). From this group of models we choose the Integrated Random Walk (IRW) trend model. The IRW trend model extends the linear regression trend line by a *flexible trend* while retaining all uncertainty information (Visser, 2004; Visser et al., 2012; Visser et al., 2015). Furthermore, the flexibility of the trend model is optimized by Maximum Likelihood (ML) optimization. The IRW model reads as:

$$y_t = \mu_t + \varepsilon_t \quad \text{and} \quad \mu_t - 2\mu_{t-1} + \mu_{t-2} = \eta_t \qquad (2)$$

where $y_t$ denotes a measurement at time t and $\mu_t$ the trend component. The terms $\eta_t$ and $\varepsilon_t$ are independent, normally distributed white noise processes with zero mean.

The Kalman filter is the ideal filter here since it yields the so-called Minimum Mean Squared Estimator (MMSE) for the trend component in the model. The Kalman filter has been applied in many fields of research and is gaining popularity in climate research recently (e.g., Hay et al., 2015). As with OLS methods, residuals - or innovations in terms of the Kalman filter - should be white noise. We will use the $N_{eff}$ correction method in case of correlated innovations (if necessary).

A third and fourth approach applies a combination of a trend model and the statistical structure of natural internal variability as derived from PiControl runs. It can be seen as a hybrid approach. To do so we have chosen the cubic spline trend model, a trend approach also applied in the AR5 (IPCC, 2013 - Box 2.2, figure 1). For a theoretical background we refer to Hastie et al. (2001) and Chandler and Scott (2011 - Section 4.1.3).

Smoothing splines are not statistical in nature and, thus, do not generate uncertainty estimates for GMST increments $\Delta\mu_{2016}$. However, uncertainty bands can be reconstructed by Monte Carlo (MC) simulations under the assumption of a given mean, variance and autocorrelation structure estimated directly from the underlying dataset (Mudelsee 2014 - section 3.3). See figure 2 for an illustration.

To steer the flexibility of the cubic spline model we studied the correlation structure of internal variability. This correlation structure can be described by an AutoRegressive Moving Average (ARMA) model as proposed by Hunt (2011) and Roberts et al. (2015). They estimated ARMA models to a range of PiControl runs. Similarly, we analyzed 41 PiControl runs with lengths varying between 200 and 1000 years. We found that variability can

reasonably be characterized by AR(1) processes where the AR(1) parameter φ varies within the range [0.0, 0.75], depending on the GCM run chosen (cf. Mudelsee, 2014 - section 2.1). In this study we have removed the lowest and highest two φ estimates yielding the range [0.28, 0.60].

We note that in some cases MA(1) or ARMA(1,1) models performed somewhat better as checked by comparing AIC values. Thus, the AR(1) model is a compromise to ease the analysis. Next to that AR(1) models are widely applied in climate research (e.g., Mudelsee, 2014).

All four trend methods are designed to smooth GMSTs for annual to decadal natural variability (forced and unforced). However, if Paris targets should be interpreted as anthropogenic warming only, we should estimate the role of decadal to centennial forcings from volcanic and solar activity as well. To estimate the role of volcanic eruptions we have extended the OLS linear trend model and the IRW trend model by adding the aerosol optical depth (AOD) index as regressor (Visser and Molenaar, 1995; Visser et al., 2015 - figure 4). The extended IRW model reads as:

$$y_t = \mu_t + \alpha x_t + \varepsilon_t \quad \text{and} \quad \mu_t - 2\mu_{t-1} + \mu_{t-2} = \eta_t \tag{3a}$$

where the variable $x_t$ stands for the inclusion of an explanatory variable (regressor). The AOD index is available from NASA for the period 1850-2016 (Sato et al, 1993; Ridley et al., 2014).

We note that if the variance of noise process $\eta_t$ in model (3a) is set to zero, the model reduces to the OLS multiple regression model with one regressor:

$$y_t = \alpha_0 + \alpha_1 t + \alpha_2 x_t + \varepsilon_t \tag{3b}$$

Thus, model (3b) is a special case of model (3a).

## 3 Results

### 3.1 Sensitivity analysis trend methods and data products

Based on the 1880-2016 GMST sample period we have evaluated trend progression values $\Delta\mu_{2016}$ from 1880 up to 2016 along with uncertainties for all datasets and trend approaches. This yields the 4-by-5 matrix shown in table 2. As for linear trends we corrected uncertainty estimates by a factor $\sqrt{(1.60/0.40)} = 2.0$, analogous to the approach chosen in IPCC (2013 - Ch. 2, Sup. Mat.) since first-order autocorrelations lie around 0.60. Table 2 shows that the trend slopes for the dataset HadCRUT4, LOTI-NASA, NOAA-Karl and Cowtan and Way are close, where the lowest slope value is for the HadCRUT4 series. This dataset has poor coverage in the Arctic, where trends are much higher than the global mean. The steepest trend is found for the Berkeley Earth series. Identical patterns are

found for the other trend models: lowest trend progression for the HadCRUT4 dataset and highest values for the Berkeley Earth dataset.

As for the IRW trend estimates - formulated in eq. (2) - we find reasonable flexible patterns which closely resemble the spline trend shown in IPCC (2013 - Ch.2: Box 2.2, figure 1b). An example for the HadCRUT4 dataset is shown in figure 3. Data, trend and uncertainties are shown in the upper panel. The trend increments $[\mu_t - \mu_{t-1}]$

and $[\mu_t - \mu_{1880}]$ are given in the middle left and right panel, respectively, along with uncertainties (cf. explanations given in Visser, 2004). The $[\mu_{2016} - \mu_{1880}]$ value with uncertainty is taken as value in table 2. The lower left panel shows the innovations or one-step-ahead predictions errors which follow from the Kalman filter formulae. The lower right panel shows the autocorrelation function (ACF). We note that a prerequisite of Kalman filtering is that the innovations - also denoted as one-step-ahead prediction errors - follow a white noise process. The ACF shows

an AR(1) value of 0.30 which is slightly significant. We applied the $N_{eff}$ correction for compensating for this the violation by applying the approach of IPCC, as we did for linear trends: uncertainty bands are corrected by a factor $\sqrt{(1.30/0.70)} = 1.3$.

As for smoothing splines we have estimated trends in GMST series such that the residual series exhibits an AR(1) process with a φ value of 0.28 and 0.60. Trend estimates based on the HadCRUT4 series are shown in figure

3. Both spline approaches show quite different trend patterns. The model shown in the upper panel of figure 4 is based on a slightly correlated noise process and - as for the IRW trend from figure 3 - closely resembles the spline trend shown in IPCC (2013 - Ch.2: Box 2.2, figure 1b). The model shown in the lower panel shows a parabolic shape. This parabolic pattern closely resembles the anthropogenic signal in GMST series as shown by IPCC (2013 - figure 10.1f), derived from 'historicalGHG' simulation runs (Forster et al., 2013).

It is interesting to note that none of the four trend methods show a sign of a 'hiatus', 'slowdown' or 'pause'. That is not surprising for the linear trend and the spline estimate with φ = 0.60 due to their stiff character. However, the IRW trend and spline with φ = 0.28 are more flexible and do not show any stabilisation pattern for recent years at all. We tested the residuals of the IRW trend model and these appear to be close to white noise (cf. lower panels of figure 2). This inference is consistent with recent findings on the hiatus (Marotzke et al., 2015; Hedemann et

al., 2017; Medhaug et al., 2017; Rahmstorf et al., 2017).

Table 2 shows that differences between trend model and dataset combinations can be considerable. The lowest $\Delta\mu_{2016}$ value is found for the HadCRUT4 dataset in combination with the IRW trend model: 0.90 ± 0.18°C (± 2σ). The highest values are found for the Berkeley Earth dataset in combination with cubic spline interpolation and φ = 0.28: 1.12 ± 0.13 °C. These two extremes reveal that the range of $\Delta\mu_{2016}$ values due to datasets and trend

models accounts for 0.22 °C. This range is somewhat lower than that due to natural variability alone. Based on 2σ limits, we find a low estimate of ± 0.12 °C, leading to a maximum range of 0.24 °C (LOTI dataset in combination with cubic spline interpolation and φ = 0.28), and a high estimate of ± 0.19 °C, leading to a maximum range of 0.38 °C (three combinations in table 2).

To quantify the role of trend methods in more detail we have averaged trend estimates over the five GMST
datasets and added it to table 2 (bottom row). It shows that the range of trend progressions is small: [0.97, 1.01]
ºC. At the other hand, if we average *over trend methods*, the variability due to datasets is found (right column of
table 1). The variability accounts for [0.92, 1.09] ºC. Clearly, variability due to GMST datasets is dominant over
specific trend approaches.


## 3.2 Trend progression derived from GCM simulations

Trend progression derived from GCMs have been analyzed in a range of studies, e.g. IPCC(2013 - Ch. 10), Forster
et al. (2013), Marotzke and Forster (2016), Mann et al. (2016) and Meehl et al. (2016). Here, we derive trend
progression since pre-industrial by taking an ensemble of 42 GCM all-forcing simulations 1861-2016. We note
that underlying models have quite different characteristics, such as climate sensitivities, various models for
greenhouse gas cycling models, cloud parametrization and aerosol forcing. However, we did not perform an
sensitivity analysis as for these factors.

Short-term forced and unforced natural variability in individual GCM simulations is smoothed by estimating
splines to each individual simulation (both for $\varphi = 0.28$ and $\varphi = 0.60$, as in figure 4). In this way we find 42 values
for $\Delta_{i,2016} \equiv y_{i,2016} - y_{i,1861}$ . Results are shown in figure 5 (based on smoothing splines with $\varphi = 0.28$). The mean
$\Delta_{2016}$ value is $1.17 \pm 0.50$ ºC $(2\sigma)$ for smoothing all 42 curves with $\varphi = 0.28$ and
$1.01 \pm 0.52$ ºC for smoothing with $\varphi = 0.60$. These values are consistent with those reported by Forster et al. (2013,
table 3).
The GCM simulations analyzed here differ from data products as for their definition of temperatures ('tas only'
versus blended temperatures). Cowtan et al. (2015) and Richards et al. (2016 - figure 1) showed that tas
temperatures differ from blended temperatures by 0.10 ºC, for the period 1860-2009. Thus, mean GCM-derived
warming estimates cover the ranges [1.00 to 1.15] ºC (tas) or [0.90 to 1.05]  ºC (blended). We note that these
ranges reasonably correspond to the range found in table 2.

## 4   Discussion

### 4.1 Uncertainty and sensitivity analysis

We make three comments concerning the robustness of the results given in section 3. First, as summarized in table
A.1 of Appendix A, a wide range of trend models exist in the literature, all with varying characteristics. The fact
that many of these methods are not statistical in nature does not limit their application in the present context: the

approach shown in figure 2 (creating surrogate GMST series by MC simulation) is also applicable to methods such as binomial filters or LOESS estimators. Therefore, we cannot rule out that the influence of trend modelling is underestimated in table 2. However, given the (i) small differences shown in the bottom row of table 2, and (ii) the wide uncertainty bands due to natural variability, we judge such an under-estimation to be relatively small.

A second comment concerns a source of uncertainty dealing with the choice for year or period that can be regarded as 'pre-industrial'. As for the analyses in section 3.1, we have chosen for the year 1880 as low end of the sample period, simply because two out of five GMST products start in 1880 (NASA and NOAA). Both NOAA and NASA reason that SST data for the pre-1880 period are too sparse (Hansen et al., 2010 - indentation [15]). The choice for 1880 is consistent with that made by IPCC (2013) as for historic trend progression (without claiming this to be 'since pre-industrial'). In section 3.2 we have chosen the year 1861 as low end of the sample period, again since simulations are available from that year onwards.

Would our results and conclusions from table 2 or figures 3 and 4 be different if the sample period would be enlarged, starting in 1400, 1720 or 1850? Strictly spoken, we cannot answer this question since we cannot extend our analyses to these starting years due to data availability. As for instrumental dataset, we could perform some analyses from 1850 onwards but GMST estimates become inaccurate for these early decades. However, estimates based on GCM simulations are given by Hawkins et al. (2017) and Schurer et al. (2017).

Hawkins et al. show that the GMST difference between the two periods 1720-1800 and 1850-1900 is small, around 0.05 ºC, lying on the edge of statistical significance. Additionally to their analysis we compared GMST mean values over three periods: 1850-1900, 1860-1880 and 1880-1900, based on the HadCRUT4 dataset. The mean values appear to be similar: -0.31 ± 0.03 °C, -0.31 ± 0.06 °C and -0.32 ± 0.05 °C, respectively (2σ limits). These differences are small if compared to the uncertainties due to natural variability, shown in table 2. These results suggest that the choice for 1720-1800, 1850-1900, 1860-1880 or 1880-1900 as 'pre-industrial' will have a small influence to the findings presented here. At the other hand, Schurer et al. show from GCM simulations that global warming is underestimated by 0.09 [0.03, 0.19] ºC if the period 1401-1800 is chosen as pre-industrial baseline (compared to the period 1850-1900). Their estimate for the influence of GHG only lies close to these estimates, in the range from 0.02 to 0.20 ºC. We conclude that recent simulations point to an underestimation of global warming if calculated relative to late nineteenth century estimates. The underestimation lies around 0.10 ºC.

A third comment deals with differences in warming definitions as mentioned in the Introduction. If the Paris targets should be interpreted as anthropogenic warming only, we should estimate these contributions as well. Clearly, the incremental estimates $\Delta\mu_{2016}$ shown in table 2 do not contain corrections for decadal to centennial natural forcings from solar and volcanic activity. To estimate the role of volcanic activity on the estimates given in table 2 we have extended the OLS linear trend and the IRW trend model with a regression component where GMST series are regressed on the OAD index shown in figure 6, following models (3a) and (3b). Results are summarized in table 3. The table shows that incremental estimates $\Delta\mu_{2016}$ are overestimated by 0.02 ºC for linear

trends and by 0.04 ℃ for IRW trends. A reason for this overestimation could be the high volcanic activity for the
period 1880-1890, containing the peak eruption of the Krakatoa).

To estimate the role of long-term solar activity we did not choose for the time-series approach above since any explanatory variable in a regression model with some long-term trend will correlate and 'explain' the long-term trend in the dependent variable (the cyclic pattern in solar radiance is not reflected in GMTs as shown by a number of studies, e.g. Schurer et al. 2017- figure S3). Therefore, we prefer to use GCM estimates to quantify the role of
solar activity.

IPCC (2013) estimates the role of solar variability to be small and on the edge of significance. Incremental solar forcing for the period 1750-2011 accounts for 2 [0, 4] % of GHG forcing (Figure SPM.5 and Box 10.2). Schurer et al. (2017 - figure S3) estimate the incremental contribution of solar forcing on GMSTs to be 0.07 [0.02, 0.12] ℃. This estimate compares the period 1850-1900 to 1990-2000. Furthermore, the long-term influence of volcanic
activity is non-significant in their simulations (their figure S2).

Next to these estimates we analyzed an ensemble of 37 GCM simulations with natural forcing only ('historicalNat'; IPCC, 2013 - figures 10.1 and 10.7; Forster et al., 2013 - fig 2). The mean curve with 2 standard errors (SEs) is shown in figure 7, along with major volcanic eruptions (eruptions with a Volcanic Explosivity Index of 5 and 6). Mean trend progression for these 37 runs accounts for 0.078 ± 0.030 ℃ (2 SE), 1861-2005.
From these inferences we conclude that the difference between total warming and anthropogenic warming lies around 0.10 ℃ with an uncertainty range of [0.0, 0.14].

### 4.2   Policy recommendation

Schurer et al. (2017) end their article with the recommendation that a consensus be reached as to what is meant by pre-industrial temperatures. In this way, the chance would be reduced of conclusions that appear contradictory being reached by different studies. Furthermore, it would allow for a more clearly defined framework for policymakers and stakeholders. We fully agree with this recommendation. However, our uncertainty and
sensitivity analysis has shown that the choice of a proper pre-industrial baseline is not the only parameter that could lead to contradictory results. Decisions around data products and GCM simulations, various time series techniques, or assumptions on warming definitions should be taken into account as well.

Here, we make the following policy proposal which aims to be a reasonable compromise. First, we propose to base GMST warming estimates on data products rather than GCM simulations. Our argumentation is that $\Delta_{2016}$
values based on GCM simulations show a wide range of warming estimates (figure 5). We note that even wider ranges are found for *absolute* GMST estimates (CMIP5 estimates for the mean GMST value over the period 1961-1990 show a range of 2.5 ℃ according to IPCC 2013 - figure 9-8). Another argument is that forcing estimates

from CMIP5 are accurate up to the year 2005 (estimates for 2006-2016 apply to approximations for GHG concentrations, with no volcanic or solar activity).

Second, since warming estimates vary as a function of the GMST data products chosen (table 2), we propose to estimate trends on the annual averages of all five data products.

Third, we found that the choice for specific trend methods plays a minor role, with largest differences between stiff and more flexible trend models. Therefore, we propose to apply a flexible and a stiff trend method and average the warming estimates found.

Fourth, two studies on the role of pre-industrial baselines have been published recently. Schurer et al. (2017) find a GHG-induced warming in the range [0.02, 0.20] ºC if the period 1401-1800 is compared to the period 1850-1900. Hawkins et al. (2017) define the period 1720-1800 as a reasonable baseline for pre-industrial and find small non-significant differences between the period 1720-1800 and 1850-1900. We choose to follow the baseline proposed by Hawkins et al. Since GMST observational data are uncertain in the pre-1880 period (sparse SST data) 395 and GMST mean values for 1850-1900 and 1880-1900 appear to be of equal size (based on the HadCRUT4 data product), we propose to analyse trend progression from 1880 onwards.

Finally, we propose to interpret global warming in the context of "Paris" as the sum of natural and anthropogenic warming, consistent with the IPCC definition of climate change. One argument for this choice is that ecological systems and human society will respond to total warming and induced shifts in climate extremes 400 *regardless of its origin*.

From these choices it follows that trend progression $\Delta_{2016}$ accounts for $1.00 \pm 0.13$ ºC (bottom row of table 2). It is interesting to compare this estimate with that published recently by Haustein et al. (2017). They find for GMST warming the incremental value 1.01 [0.87, 1.22] ºC which is close to our findings. This is remarkable since their estimate is based on another approach and quite different assumptions.


## 5 Conclusions

We have addressed the issue of signal progression of GMST in relation to the GMST targets agreed upon in Paris 410 in December 2015. Although these targets are clearly defined – avoiding increments of 1.5 and 2.0 °C – there remain a number of (scientific) questions unanswered in the agreement. We have identified five aspects of the accord which hamper an exact quantification of GMST progression: (i) the use of instrumental data and trend methods versus GCM-derived progression, (ii) the role of varying datasets, (iii) the role of varying trend methods, (iv) the role of varying choices for pre-industrial and (v) the role of warming definitions. Since there is no 'true' or

'best' approach (Visser et al., 2015), we have chosen to perform an uncertainty and sensitivity on GMST progression as propagated by Saltelli et al. (2004) and related articles. This allows us to test the robustness of various trend progression claims.

*Approaches based on instrumental data.* We find that trend values for GMST progression 1880-2016 vary considerably, from 0.90 ºC (HadCRUT4 dataset in combination with the IRW trend model) to 1.12 ºC (Berkeley
Earth dataset in combination with cubic spline interpolation and $\varphi = 0.28$). The two extremes reveal that the range of $\Delta\mu_{2016}$ values due to datasets and trend models accounts for 0.22 °C. This range is smaller than that due to natural variability alone. Based on $2\sigma$ limits, we find a low estimate of 0.24 °C (LOTI dataset in combination with cubic spline interpolation and $\varphi = 0.28$) and a high estimate of 0.38 °C (three combinations in table 2). Furthermore, variability due to various GMST products dominates the variability due to specific trend approaches.

*Approaches based on GCMs*. We find that mean trend progressions lie within the range of estimates from instrumental data. However, the uncertainty bands for 42 simulations are much wider than those derived from instrumental trend estimates. Here, GCM variability stems from a wide range of modeling assumptions such as climate sensitivities, cloud parameterization and aerosol forcing (e.g., IPCC, 2013 - Ch. 9), in addition to natural variability.

*The choice of a pre-industrial period*. Recent studies have shown that GHG warming prior to 1880 or 1850 cannot be neglected. Schurer et al. (2017) estimate that early warming (1401-1800 compared to 1850-1900) accounts for 0.09 [0.03, 0.19] ºC. The role of solar and volcanic activity is minimal in this comparison.

*Interpretation of Paris targets as being 'total warming' or 'anthropogenic warming only'*. We find that the role of solar and volcanic activity is small on centennial scale. This contribution lies around 0.10 ºC (0.03 ºC from
volcanic activity and 0.07 ºC from solar activity; cf. section 4.1 for an explanation).

*Hiatus*. As a side result of our trend analyses we note that no signs of an 'hiatus', 'slowdown' or 'pause' can be discerned in GMST trend progression. This inference is consistent with recent findings (Marotzke et al. 2015, Hedemann et al. 2017, Medhaug et al. 2017, Rahmstorf et al. 2017).

*Policy recommendation*. Schurer et al. (2017) recommend that a consensus be reached as to what is meant by
pre-industrial temperatures. Our analysis shows that other sources of uncertainties should be taken into account as well. If not, contradictory results will appear in different studies with direct consequences for $CO_2$ reductions to hold GMSTs below the Paris targets. Our proposal shows a GMST progression $\Delta_{2016}$ of 1.00 ºC.


**Table 1.** Summary of observational datasets used in this study. Descriptions of interpolation schemes are only short indications. Details are given in the references.

| GMST dataset | Land product | SST product | Interpolation method | Period | Key references | Website |
|---|---|---|---|---|---|---|
| **HadCRUT4, CRU version 4.6** | CRUTEM4.6 | HadSST3.1 | Spatial correlation structures within (land) and between (sea) grid boxes | 1850-2016 | Morice et al. (2012) | here |
| **HadCRUT4, Cowtan and Way version 2.0** | CRUTEM4.5 | HadSST3.1 | Infill procedure by Kriging and a hybrid method guided by UAH satellite data | 1850-2016 | Cowtan and Way (2014) | here |
| **LOTI series, NASA-GISSTEMP** | GHCN v3 and Antarctic (SCAR) data | ERSSTv4 | Weighted averages of anomalies for all stations within 1200 km of that point. | 1880-2016 | Hansen et al. (2010) | here |
| **NOAA, Karl et al.** | GHCN-M v3.3 | ERSSTv4 | Grid-box averaging | 1880-2016 | Smith et al. (2008), Karl et al. (2015) | here |
| **Berkeley Earth Project** | GHCN database (modified version) | HadSST3 (modified version) | Infill procedure by Kriging | 1850-2016 | Rohde et al. (2013) | here |



**Table 2.** Trend increments $\Delta\mu_{2016}$ along with $2\sigma$ confidence limits. Increments are given for the five GMST series given in table 1, and the four trend approaches proposed in section 2.2.

| GMST dataset | GMST progression $\Delta\mu_{2016}$ with $2\sigma$ confidence limits ($^\circ$C) | | | | |
|---|---|---|---|---|---|
| | OLS linear trend | IRW trend | Spline with $\varphi=0.28$ | Spline with $\varphi=0.60$ | Mean progression |
| HadCRUT4, CRU | 0.90 ($\pm$ 0.18) | 0.93 ($\pm$ 0.17) | 0.94 ($\pm$ 0.12) | 0.92 ($\pm$ 0.14) | 0.92 |
| HadCRUT4, Cowtan and Way | 0.96 ($\pm$ 0.17) | 1.06 ($\pm$ 0.17) | 1.06 ($\pm$ 0.12) | 0.98 ($\pm$ 0.15) | 1.02 |
| LOTI series, NASA | 0.98 ($\pm$ 0.19) | 1.02 ($\pm$ 0.18) | 1.01 ($\pm$ 0.12) | 0.99 ($\pm$ 0.14) | 1.00 |
| NOAA, Karl et al. | 0.95 ($\pm$ 0.19) | 0.96 ($\pm$ 0.19) | 0.94 ($\pm$ 0.14) | 0.95 ($\pm$ 0.14) | 0.95 |
| Berkeley Earth Project | 1.04 ($\pm$ 0.17) | 1.12 ($\pm$ 0.17) | 1.12 ($\pm$ 0.13) | 1.06 ($\pm$ 0.14) | 1.09 |
| Mean progression | 0.97 | 1.02 | 1.01 | 0.98 | 1.00 |

**Table 3.** GMST progression 1880-2016 with and without correction for volcanic activity (cf. figure 6).

| GMST dataset | GMST progression $\Delta\mu_{2016}$ with 2σ confidence limits (°C) | | | |
|---|---|---|---|---|
| | OLS linear trend | IRW trend | OLS linear trend with regression on AOD | IRW trend with regression on AOD |
| HadCRUT4, CRU | 0.90 (± 0.18) | 0.93 (± 0.17) | 0.89 (± 0.18) | 0.89 (± 0.17) |
| HadCRUT4, Cowtan and Way | 0.96 (± 0.17) | 1.06 (± 0.17) | 0.94 (± 0.18) | 1.01 (± 0.17) |
| LOTI series, NASA | 0.98 (± 0.19) | 1.02 (± 0.18) | 0.97 (± 0.20) | 0.98 (± 0.18) |
| NOAA, Karl et al. | 0.95 (± 0.19) | 0.96 (± 0.19) | 0.94 (± 0.20) | 0.95 (± 0.19) |
| Berkeley Earth Project | 1.04 (± 0.17) | 1.12 (± 0.17) | 1.02 (± 0.17) | 1.07 (± 0.17) |
| Mean progression | 0.97 | 1.02 | 0.95 | 0.98 |



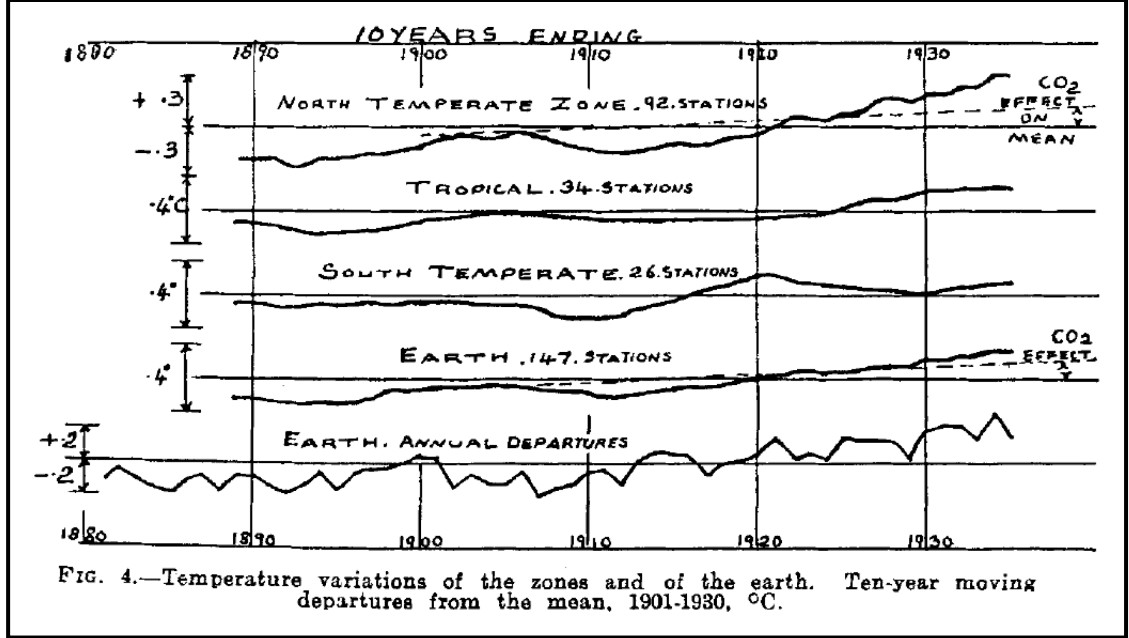

FIG. 4.—Temperature variations of the zones and of the earth. Ten-year moving departures from the mean, 1901-1930, °C.

**Figure 1.** Graph taken from Callendar (1938). The fourth curve represents his GMST series, based on temperature data of 147 stations. To highlight smooth changes over time he used moving averages with a window of 10 years. It is interesting to note that he also addressed the specific effect of $CO_2$ emissions on global temperatures (dashed lines).





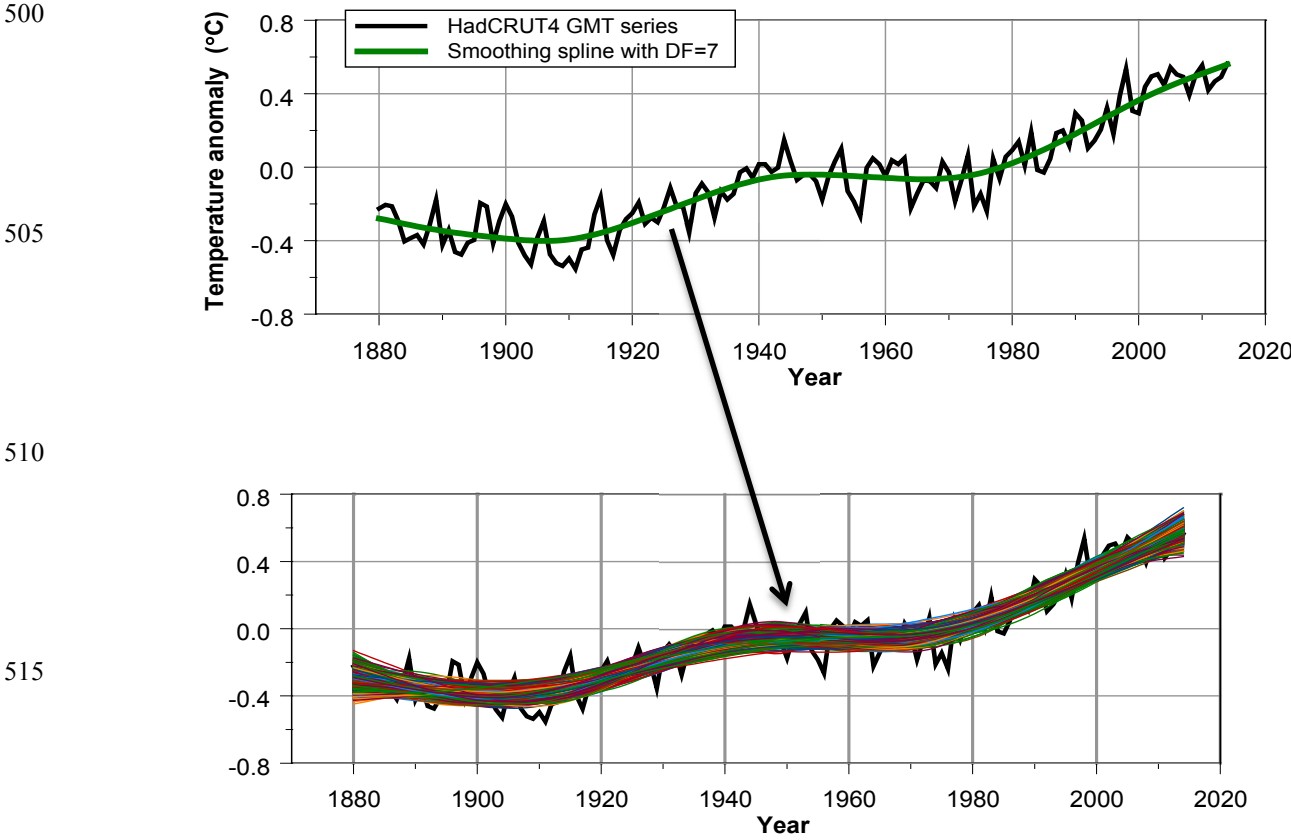




**Figure 2.** Construction of 1000 surrogate trend series by MC simulation, based on cubic splines. The AR(1) parameter estimated on the residuals of the spline model in the upper panel, accounts for 0.28. A surrogate GMST series $\hat{y}_{i,t}$ is formed by simulating a new residual series $r_{i,t}$ based on the AR(1) process with $\varphi = 0.28$, and adding it to the estimated spline (green line upper panel). Then, a spline trend $\mu_{i,t}$ is estimated for each surrogate $\hat{y}_{i,t}$. As an
illustration we have plotted 1000 of such trends $\mu_{1,t}$ , ... , $\mu_{1000,t}$ in the lower panel. Now, confidence limits can be estimated for any $\mu_t$ based on the values $\mu_{1,t}$ , .... , $\mu_{1000,t}$ . These confidence limits can be based on standard deviations or percentiles. Similarly, confidence limits can be calculated for the increment $[\mu_{2016} - \mu_{1880}]$, based on the values $[\mu_{1,2014} - \mu_{1,1880}]$ ,...... , $[\mu_{1000,2014} - \mu_{1000,1880}]$ (Mudelsee, 2014 - sections 3.3.3 and 3.4).

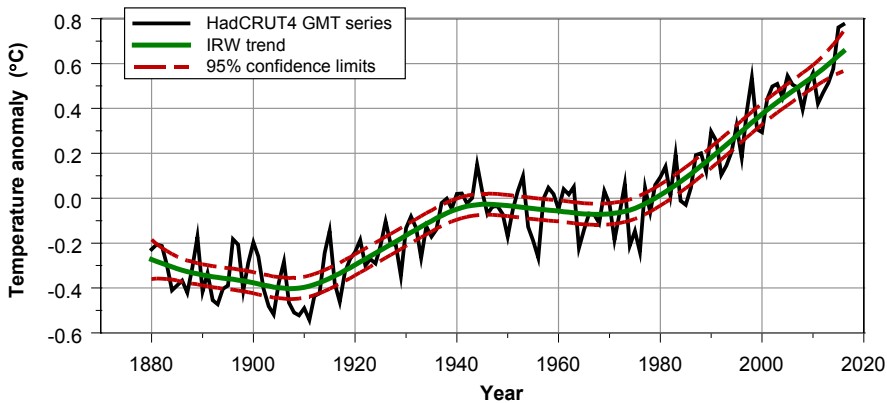

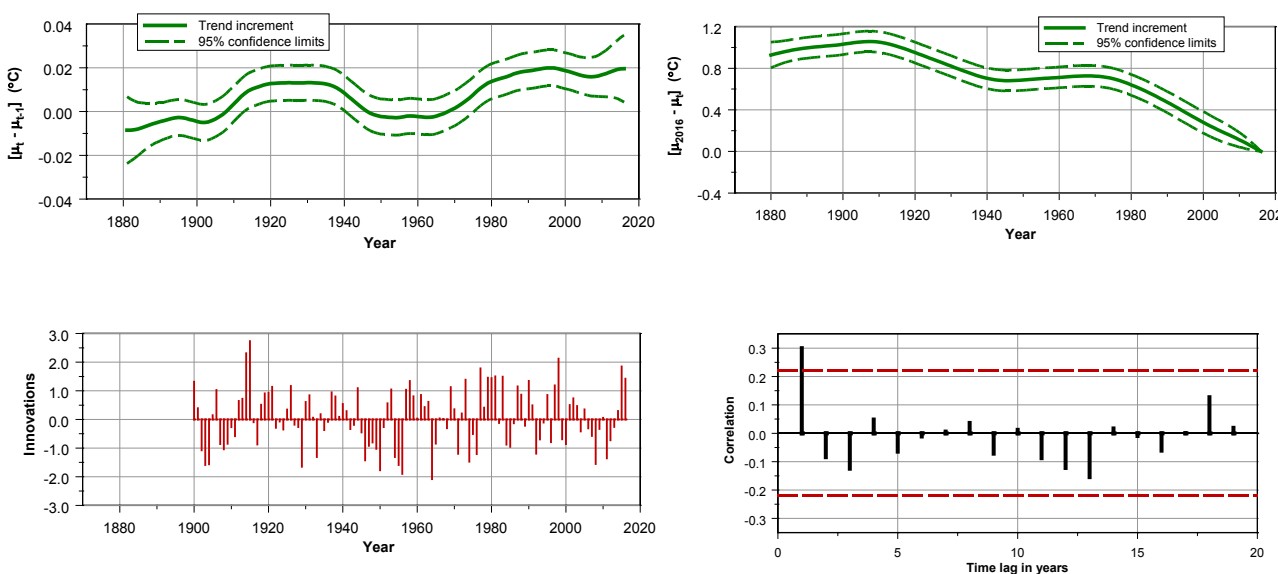


**Figure 3.** Results for the IRW trend model as applied to the HadCRUT4 series. Period: 1880-2016. The upper panel shows the trend (green line) along with 95% confidence limits (red dashed lines). The trend increments $[\mu_t - \mu_{t-1}]$ are given in the middle left panel along with uncertainties. Idem the $[\mu_t - \mu_{1880}]$ values in the middle right panel. The lower left panel shows the innovations or one-step-ahead predictions errors which follow from the

Kalman filter formulae. The lower right panel shows the autocorrelation function (ACF).

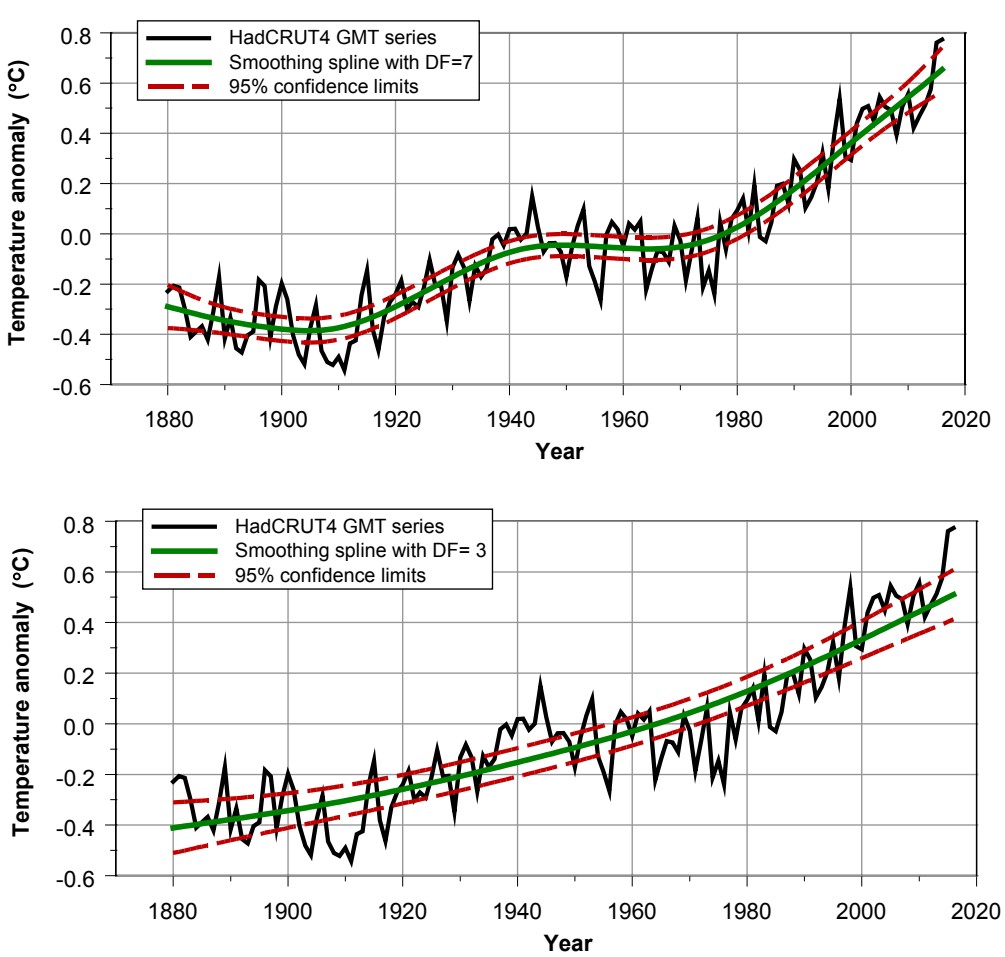

**Figure 4.** Two smoothing spline estimates for the HadCRUT4 GMST series, with uncertainties generated by MC
simulation. All confidence limits are based on 1000 surrogate GMST series following the approach set out in
Mudelsee (2014 - Section 3.3.3). Upper panel: AR(1) parameter chosen as $\varphi = 0.28$ (equivalent to 7 degrees of
freedom), the low end of $\varphi$ values within CMIP5 PiControl runs. Lower panel: AR(1) parameter chosen as
$\varphi = 0.60$, the high end of $\varphi$ values (DF=3).


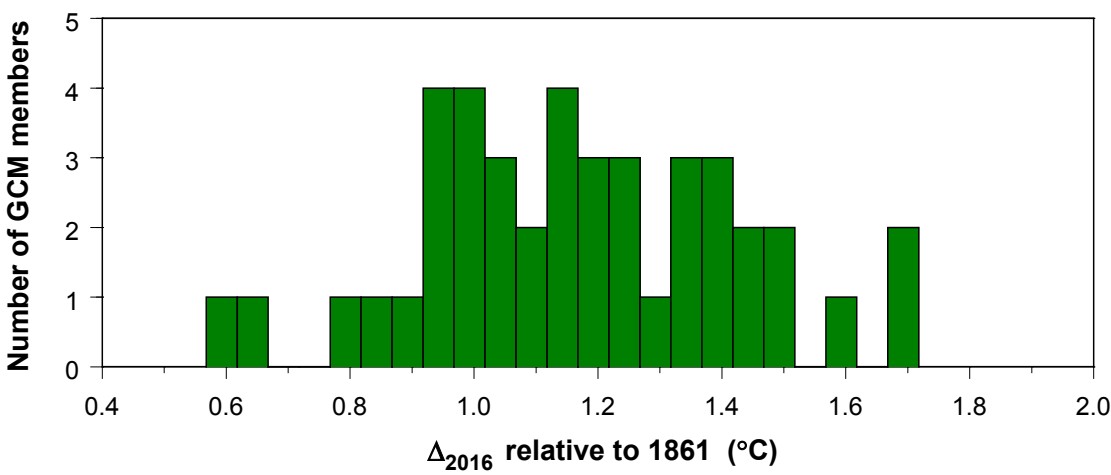

**Figure 5.** Histogram based on 42 GCM $\Delta_{i,2016}$ values, relative to 1861. Mean value is $1.17 \pm 0.50$ ºC ($2\sigma$). Individual GCM curves were smoothed by splines where the AR(1) parameter is chosen as $\varphi = 0.28$ (equivalent to 7 degrees of freedom), the low end of $\varphi$ values within CMIP5 PiControl runs.

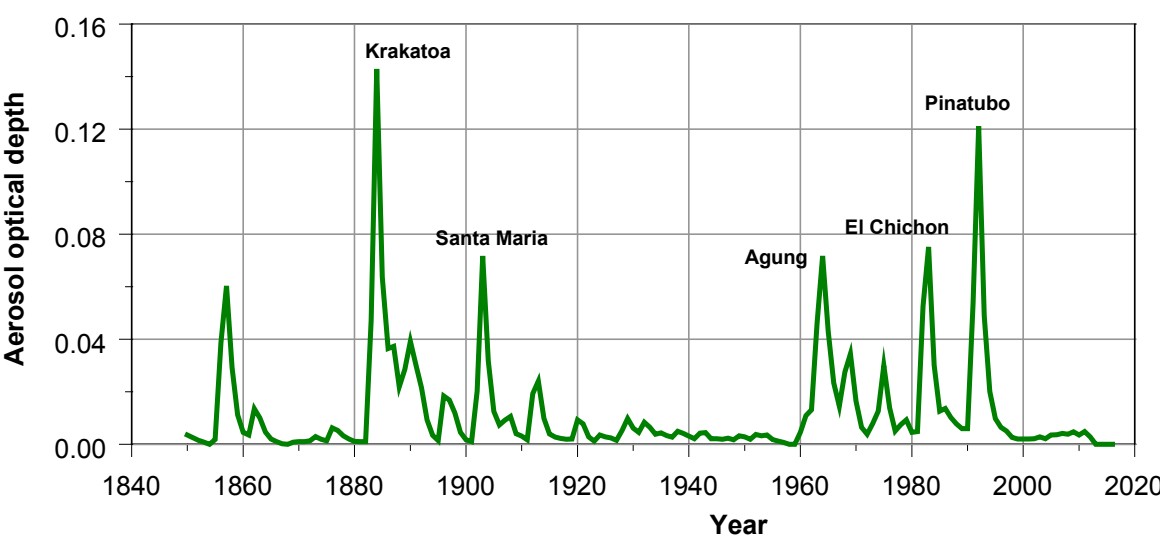


**Figure 6.** The AOD index series as introduced by Sato et al. (1993). Period is 1850-2016.


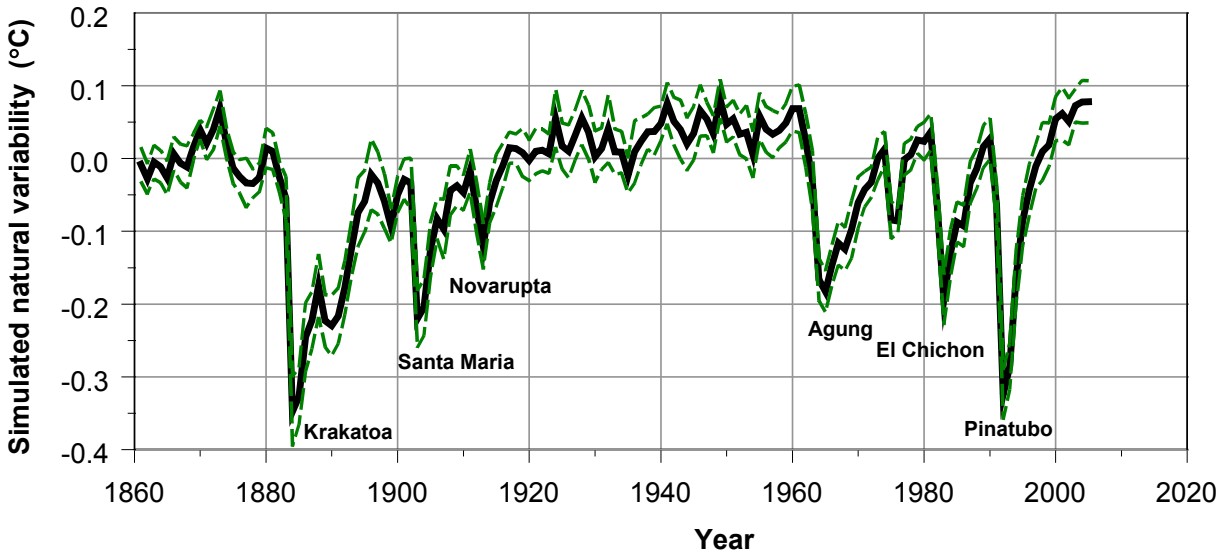

**Figure 7.** Natural variability based on 37 GCM simulations. Shown are mean values along with 2 standard errors. Period is 1861-2005.

**Code availability.** IRW trends have been estimated by the TrendSpotter software. This software package is freely available from the first author. Splines have been estimated by the statistical package S-Plus, version 8.2. The scripts which are highly similar to R, are available from the first author.

**Data availability.** All five GMST datasets are open access and have been downloaded from the authors websites. All CMIP5 runs named in Section 2.1 were downloaded from the KNMI Climate Explorer website with one member per model (Trouet and Van Oldenborgh, 2013). The names of individual GCMs can be found there as well. Please see https://climexp.knmi.nl/cmip5_indices.cgi?id=someone@somewhere . Data used for the graphical presentations in this article can be gained from the first author.

**Competing interests.** The authors declare that they have no conflict of interest.

**Acknowledgments.** We thank Geert Jan van Oldenborgh (KNMI, Climate Explorer) for thorough comments on an early version of this text. Furthermore, we thank Peter Thorne (Maynooth University), an anonymous reviewer and Lenny Smith (The London School of Economics and Political Science) for important comments on the manuscript.

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

**Appendix A   An overview of trend methods, applied to GMST observational data**

In our study we have selected trend models which not only estimate a trend over time but also yield uncertainties for trend increments. However, this requirement appears to limit our model choices considerably. First, many methods are not statistical in nature, such as moving averages (Hansen et al., 2010; Smith et al., 2015; Fyfe et al., 2016), binomial filters (Morice et al., 2012), wavelets with scale dependencies (Lin and Franzke, 2015), EEMD decomposition (Wei et al., 2015; Yao et al., 2015) or linear trends based on stair-step averages with variable lengths

(De Saedeleer, 2016). A historic example is given in figure 1, based on the work of Callender (1938).

Next to that, a number of methods do not generate estimates at the beginning and ending of the GMST series due to the dependence on 'windows'. Examples are moving averages, OLS linear trends with moving windows (Risbey et al., 2015; Marotzke and Forster, 2015) and the staircase approach by De Saedeleer (2016).

Trend models applied to GMST datasets can be categorized methodological into three groups:

- Empirical models. These are trend models which are in principle data-based and may be steered by qualitative physical insights, such as the choice of a fixed window in combination with moving averages (Easterling and Wehner 2009; Hansen et al., 2010; Cowtan and Way, 2014; Roberts et al., 2015). Other trend models are OLS linear trends with varying sample periods (IPCC 2013 - Box 2.2,

figure 1a; Karl et al., 2015; Rajaratnam et al., 2015), linear trends with change points (Cahill et al., 2015), binomial filters (Morice et al., 2012), splines (IPCC, 2013 - Box2.2, figure b), EEMD decomposition (Wei et al., 2015; Yao et al., 2015), structural time series models (Visser and Molenaar, 1995; Mills, 2006, 2010) and long-memory trend models (Lennartz and Bunde, 2009; Rea et al., 2011).

- Semi-empirical methods with stationary regressors. These methods are also data-based but physics may

enter trend estimates by adding stationary climate indices in the context of regression models. An example is given by Forster and Rahmstorf (2011) who apply a linear regression model with three regressors (MEI, AOD and TSI). Other references are Visser and Molenaar (1995), Yao et al. (2015) and Trenberth (2015).

- Semi-empirical methods with non-stationary regressors. These models differ from semi-empirical

models in that non-stationary regressors are used as well, such as global $CO_2$ emissions. Typical examples are given by Imbers et al. (2013) and Hawkins et al. (2017). An example where GMST data are treated *as regressor* to model global sea levels, has been given by Rahmstorf (2007).

A detailed description of methods is given in table A.1. For background information please see Chandler and

Scott (2011), Mudelsee (2014) and Visser et al. (2015).

From the range of available trend methods we selected trend methods from the group of empirical models and semi-empirical models where our main selection criterion is that models contain full uncertainty information for trend estimates and trend increments. Based on this criterion we selected models (4), (8), (16), (19) and (21). As for model (8) we explained the construction of uncertainties in figure 2.

Furthermore, we decided not to use models from the semi-empirical approaches with non-stationary regressors. First, there is a danger of finding associations rather than causal relations since any two series with a long-term trend correlate high, whatever their origin (Nuzzo, 2014). Second, relations in the climate system are (highly) non-linear and we prefer to rely on GCM simulations rather than forcing indicators for GHGs, aerosols or solar activity which serve as regressors in a multiple regression model. Thus, we prefer the models named in table A.1 under the heading 'Semi-empirical approaches, stationary regressors' over 'Semi-empirical approaches, non-stationary regressors'.







**Table A.1.** Summary of three groups of modeling approaches to global mean temperatures: (i) empirical, (ii) semi-empirical with stationary regressors, and (iii) semi-empirical with non-stationary regressors. In the fourth column the presence of uncertainties for rates of change is given ([$\mu_t$ - $\mu_s$] ± ?). The term 'not explicitly' means that uncertainties could be calculated in principle but not shown by the author(s).

| | Empirical approaches | | [$\mu_t$ - $\mu_s$] ± ? |
|---|---|---|---|
| 1 | Decadal aggregation, no trend | Callendar (1938 - figure SM.1), IPCC (2013 - figure SPM.1a & figure 2.19) | no |
| 2 | Moving averages with prescribed window length (varying from 5 to 50 years) | Callendar (1938), Easterling and Wehner (2009), Hansen et al. (2010, Figure 9), Kokic et al. (2014), Cowtan and Way (2014), Roberts et al. (2015) Smith et al. (2015), Fyfe et al. (2016) | no |
| 3 | OLS linear trends, with various corrections for correlated noise | Rajaratnam et al. (2015) | yes |
| 4 | OLS linear trends for varying sample periods, with corrections for correlated noise | IPCC (2013 - Ch.2: Box 2.2, figure 1a), Karl et al. (2015), this study | yes |
| 5 | OLS linear trend with moving windows | Risbey et al. (2014), Marotzke and Forster (2015) | only for [$\mu_t$ - $\mu_{t-1}$] |
| 6 | Linear trends with change points (CP) | Cahill et al. (2015), Rahmstorf et al. (2017) | not explicitly |
| 7 | Linear trends, based on stairstep averages with variable lengths | De Saedeleer (2016) | yes, by color graphs |
| 8 | Splines with Monte Carlo simulation | IPCC (2013 - Ch.2: Box 2.2, figure 1b), this study (with CMIP5-derived AR(1) noise) | yes |
| 9 | 21-term binomial filter | Morice et al. (2012) | no |
| 10 | Hodrick-Prescott and Butterworth low-pass filters | Mills (2006) | no |
| 11 | Smooth transition trends | Mills (2006) | no |
| 12 | Adaptive filtering with padding | Mann (2008) | no |

| 13 | Wavelets with scale-dependencies | Lin and Franzke (2015) | no |
|----|----------------------------------|------------------------|-----|
| 14 | EEMD decomposition | Wei et al. (2015), Yao et al. (2015) | no |
| 15 | ARIMA decomposition | Mills (2006) | no |
| 16 | IRW trend model, part of the STM group of models | Visser and Molenaar (1995), Mills (2006, 2010), model (2) of this study model | yes |
| 17 | Long memory trend models | Lennartz and Bunde (2009), Rea et al. (2011) | no |
| **Semi-empirical approaches, stationary regressors** | | | |
| 18 | Linear for selected PDO regimes | Trenberth (2015) | no |
| 19 | Multiple regression models with linear trend, aerosols and solar | Forster and Rahmstorf (2011), model (3b) of this study | yes |
| 20 | EEMD decomposition with correlations PDO and AMO | Yao et al. (2015) | no |
| 21 | STMs with regressors | Visser and Molenaar (1995), model (3a) of this study | yes |
| **Semi-empirical approaches, non-stationary regressors** | | | |
| 22 | Regression models with GHGs, SOI, TSI, volcanic, ARMA noise | Kokic et al. (2014) | not explicitly |
| 23 | Cointegration, ARIMA, trend breaks, RF, GHGs | Kaufmann et al. (2006, 2013) | not explicitly |
| 24 | Regression models with ENSO, AMO, GHG, solar, aerosols and AR(1) noise | Imbers et al. (2013), reprinted in IPCC (2013 - Ch. 10) | not explicitly |
| 25 | Regression models with forcings from GHGs, aerosols, solar activity, volcanic activity and Nino3.4 as regressors | Hawkins et al. (2017, their approach 1) | yes |
| 26 | Scaling model with local temperature series as regressors (CET, De Bilt) | Hawkins et al. (2017, their approach 3) | yes |
| 27 | Regression model with temperature responses to human-induced forcings and natural drivers as explanatory variables. Various GMST observational datasets serve as dependent variable. | Otto et al. (2015), Haustein et al. (2017) | yes |