# Peer review of "Signal detection in global mean temperatures after "Paris": an uncertainty and sensitivity analysis"

_Climate of the Past, 2017_

## Referee Comment (RC1) · P. Thorne (Referee) · 24 Jul 2017

**Review of "Signal detection in global mean temperatures after "Paris": an uncertainty and sensitivity analysis**

**Visser et al., submitted to Clim. Past Discuss**.

Visser et al perform a comprehensive timeseries analysis of Global Mean Surface Temperatures (commonly referred to as GMSTs although for some reason referred to by the authors as GMTs). The rationale given is to inform the COP negotiations, which is a laudable aim. There is a significant amount of analysis undertaken that yields potentially useful and actionable information. As such, scientifically, this constitutes a publishable work.

**Major points**

There is a question whether the paper content is in scope of the journal as it deals exclusively with the recent past and with instrumental records. The decision of in-scope or not is one for the assigned editor and the broader editorial team to take a view on. I merely flag it here.

I find the hook to pre-industrial tenuous given that the authors make no attempt to estimate a true pre-industrial based value. They would be better, in my view, to state that they are making an estimate relative to the late 19$^{th}$ Century / early global instrumental record. This would be a fairer reflection of what is actually done and consistent with e.g. IPCC AR5 which deliberately avoided in the published version implying that 1850-1900 constituted pre-industrial as noted in Hawkins et al. Indeed, the final plenary of the WG1 involved a long discussion that I was personally involved in around the topic whereby the parties agreed that pre-industrial was earlier than 1850. It would be unwise, in my view, for the authors to reopen this issue. I note in a couple of places that there are phrases which could imply IPCC used 1850-1900 as pre-industrial, and they did not. Such implications absolutely must be avoided outright in any resubmission.

The discussion linking their work to the pre-industrial era would be far better being given exclusively in the Discussion section and, to my view, the authors should remove allusions to providing an estimate relative to pre-industrial earlier than this. Bottom line: They either should estimate relative to true-pre-industrial or be honest with respect to what they are estimating relative to for the paper to be acceptable. As I see it there is no rigorous attempt to estimate changes since pre-industrial. Rather, there is a rigorous attempt to estimate it since 1880 which in itself is useful and valuable. The authors should be honest in this regard and not oversell their work by claiming its an estimate relative to pre-industrial when it demonstrably is not.

I also find the Section at the end of the paper alluding to RCPs and end of Century to be out of scope and a distraction. It should either form an integral part of the paper integrated throughout or be dropped. Given journal scope I would lean heavily toward its removal. The year 2100 is not in the past (at least yet)!

Finally, given the authors apparent desire to explore uncertainty I find the omission of the JMA observational analysis and the NOAA 20CR product odd. I could see a case for omission

of 20CR, but I see no logical case why the JMA analysis should be omitted here as it has the same non-peer-reviewed basis as e.g. the Berkeley global (not land, but global) estimate. JMA uses a fundamentally distinct set of SSTs and so would better span uncertainty that the authors lament in Section 2.1.

**More minor points**

I have a number of further comments, suggestions and requests which I refer to in the order they arise chronologically below:

1. Line 19 per above remove 'and what is 'pre-industrial'?' as you make no attempt to robustly address that question.

2. Line 47 GMTs have …. So following the 21$^{st}$

3. Lines 53-56 do not reflect the IPCC approach. This was not an attempt to inform on post-pre-industrial changes and it did not involve expert judgement. Rather the stated range is the range of available estimates and their uncertainties after correcting for AR(1) and using OLS. The text here significantly overcomplicates both what was done and why. As the IPCC author who undertook the lead on this analysis I can assure the authors it was not as complicated as they imply here. This should be revised to reflect the actual process.

4. If retained (and note earlier major suggestion to move this to discussion) line 55 forwards should constitute the beginning of the paragraph currently starting line 57.

5. Line 73 or similar do not

6. Line 73. Reader will ask so what? You need to be explicit that the approach limitations matter in a period of rapid change.

7. Line 76 progression to specific (remove allusion to pre-industrial per major comment)

8. Line 88 the main one being

9. Lines 108 to 121 omit the by far largest overlap of all in that the NOAA and NASA products are based on identical underlying land and ocean datasets differing solely in the applied post-processing. This needs to be acknowledged for this discussion to be acceptable. More generally this discussion is incomplete. It needs to be expanded and may be better if supported by a table.

10. Feels odd not to discuss and cite Cowtan et al at lines 125-127

11. Lines 155-157. First please clarify whether the AR1 factors are calculated on the annual series. This is important information that is being omitted. Secondly, even at annual scales the AR(1) is primarily an artefact of variability and not forcing so the

assertion here is wrong as you note in lines 175-177. Your two cheek-to-jowl statements here cannot both be right. The AR arising from variability is the correct one here. Year-to-year autocorrelation does not arise mainly due to forcing.

12. Line 202 you should clarify what the implications of ignoring this are or, preferably, perform the extra work necessary for its inclusion. Presumably the impact would be artificially reduced uncertainty ranges? In which case is it really safe to ignore this issue? I'm not entirely convinced and would suggest that extra work leading to its inclusion is instead warranted. Even if it ends up showing no change it would make the piece more robust. As you yourselves state the effect is statistically significant, in which case it really should be included.

13. Lines 224-227. This is true a. for this particular period and b. this particular small (and non-independent as noted in Section 2.1) draw from the broad range of plausible means by which to estimate historical changes in GMT. Hence I believe this statement oversimplifies the issues and as a result is more confident than is, in reality, warranted. The findings do not have the universality implied here and may not even be true if we instead had a further draw from the sample of plausible approaches to estimating GMTs from observations. Here, JMA's inclusion may fundamentally alter this finding which would imply non-robustness.

---

## Referee Comment (RC2) · Anonymous Referee #2 · 4 Aug 2017

This study considers the question of estimating by how much global temperatures have changed since 'pre-industrial' times, assessing the uncertainty in different trend models and due to different global temperature datasets. The analysis is interesting, though the results are not too surprising. However, I have some major concerns:

1) Framing: the authors emphasise repeatedly that they are estimating changes since a particular baseline and implying that this is what the Paris agreement meant by 'pre-industrial'. This is not the case - the introduction of Hawkins et al. (which the authors cite) discusses this issue at length. In addition, Schurer et al. (2017, NCC) was very recently published, highlighting again that there was likely some additional warming due to anthropogenic factors before 1850. The authors may also like to examine Otto et al. (2015) for an alternative approach to estimating the warming since the 19th

century. The text in the discussion on this topic is appropriate however.

2) Terminology: some of the phrasing is very confusing when referring to and/or distinguishing between natural *forced* variability (volcanic, solar) and internal *unforced* variability. These terms are sometimes mixed and it's not always clear what the authors mean. For example, in the abstract (and L86) the authors claim the models are corrected for natural variability, when they mean the forced component, but the introduction uses natural variability to mean both forced and unforced variations. On L133, the authors refer to the 'historicalNat' runs 'for natural unforced variability', which is not true - those runs include both natural forced and internal unforced variations as the next sentence correctly states. Variability is also used for the spread or range between different estimates, adding further confusion. The authors should carefully check each use of this type of phrasing and make it far more precise.

3) GCM analysis: the 106 members used cannot be 'one per model' as there were not that many models in CMIP5. It's not clear what the authors have used here - there must be more than one historical part of the runs for some of the models. There are also 43 piControls on Climate Explorer, and very few are less than 200 years, not only the 20 that the authors have used - why have they not used the others? Also, in section 3.2, the authors could use the AR(1) value from each model's own control run to fit a spline to the historical run of that same model, rather than assume the same across every model. Also, how has the correction for natural forcings been applied (L250)? Has the mean across the historicalNat runs been subtracted from each historical run? If so, this is inconsistent as the response to volcanic eruptions varies significantly across models.

Smaller points:

L47 - 21th -> 21st

L56 - this uncertainty does not include expert judgement

L87 - this sentence could be read to imply that GCMs should have 'priority' over the observations for answering the question of how much the surface has warmed, but I don't think the authors mean that?

L146 - I'm not sure the UNFCCC would suggest that pre-industrial should be defined when a particular dataset happens to begin?

L163-4 - do you need all five of those references to the lead author's previous papers?

L191 - this not appropriate for a scientific analysis

L216 - delete 'is'?

L220 & L307 - three 'variability' in the same sentence, all referring to slightly different things?! The first is a 'range'? The sentences following this are also not clear.

L279 - not sure this is quite true - there is a signal of these eruptions in the observations but it is probably weaker than the GCMs suggest. But I agree this is probably partly a coverage issue.

L288 - most of the time, the models are not tuned to the trend, but are tuned on the mean present-day climate state.

L303 - is 'best guess' the appropriate term?

L315 - it would be useful to use other historical ensemble members to check this statement (where they exist), as they provide another estimate of the change in temperature for the same model.

References:

Otto et al. (2015). Embracing uncertainty in climate change policy. Nature Climate Change, doi:10.1038/nclimate2716

Schurer et al. (2017). Importance of the pre-industrial baseline in determining the likelihood of exceeding the Paris limits. Nature Climate Change, doi: 10.1038/nclimate3345

---

## Short Comment (SC1) · 31 Aug 2017

General Comments

Visser et al (2017) provide an interesting and insightful discussion of signal detection in global mean temperature (GMT), focusing on the 1.5 degree target of the Paris Agreement of 2015. This paper could be made more informative by further consideration of three topics: (1) clarifying what is meant by "signal" and by "noise", and more specifically how (whether) natural variability can be "corrected for" in an evolving nonlinear system, (2) implications of using CMIP5 models, given that those models display a wide range of values for today's GMT , and (c) a cleaner definition of how one would detect failure to stay "well below" a temperature target, or to exceed it. These points

are expanded upon below.

Specific Comments

"Natural variability" is said to be a dominant source of uncertainty which has been "corrected for" (24). Although discussions of a climate signal coming "out of the noise" are common, the notions underlying the distinction between signal and noise in the climate context is unclear; it is not the traditional distinction of observational noise superimposed on a imprecisely measured but well-defined signal. Superposition can only be assumed in nonlinear systems given purely observational noise that has no impact on the system: natural variability, internal variability and the like alter the dynamics, and thus the "signal" itself, if such a separation exists (Smith (2001,2002)). A more appropriate conceptualization in nonlinear systems is found in consideration of an ensemble of systems each subject to a common driving and independent realizations of the relevant noise. In this case, the ensemble median would provide a well-defined signal while the distribution about it would capture the effects of noise processes. This view is of limited utility in climate science, where there is only one realization (the Earth): particular realizations need not reflect the (unobservable, non-empirical) "signal"; indeed they can diverge arbitrarily far from it. So in no sense can one expect "the" signal to emerge from the noise, given observations of a single realization. While vague appeals to something somewhat reminiscent of an adiabatic change in thermodynamics may be voiced, clear clarification of the meaning of signal and noise in the climate context would be of value.

In short: it would be useful to clarify how "natural variability" and "internal variability" might be isolated in the case of a complicated, nonlinear, evolving planetary system. How are we to make sense of the traditional notions of "signal" and "noise" given that the "noise" is not mere observational noise but actually a component of the system dynamics, and given that in nonlinear systems we cannot appeal to a principle of superposition of solutions (Smith, 2002).

It is also worth noting that the statistics community and the physical science community often hold very different notions of what a trend is: for the first, it is a statistically-consistent combination of two well-defined models (the trend model and the noise model), while for the second it is merely a systematic, often obvious drift. Statisticians require, and quantify, consistency between these two components, and reject identification of a trend if that consistency is lacking. Physical scientists often require the observations to look trendy, and the ability to reject simple statistical models given the data, when those models are known by construction not to admit a trend. The second bar is much lower.

The claim that modelling groups "have not been very successful in tuning to the observed trend" (299) suggests some knowledge as to how large the spread would be in the absence of each group knowing the observed trend (aiming for the same target). It has been argued elsewhere that knowledge of such spread would be very useful to have if, perhaps, impractical to obtain. .

Visser et al (2017) state that "mean progression derived from GCM-based GMTs appear to lie within the range of the trend-dataset combinations" (311).It would be interesting to see the variations among individual CMIP5 simulations (not the mean over them, but their distribution). The IPCC AR5 reports that variations in the global mean temperature of today's CMIP5 GCMs have a range exceeding 2.5 degrees (see right side axis labels of Figure 9-08 of Flato et al (2013)); what are the implications of our best models showing a range of GMT almost twice the 1.5 degree target? Physical and biological processes are driven by actual temperature, not anomalies. Given the current (limited) level of realism in these models, and the fact there is a great deal more in them than their basis in physical understanding, the authors might wish to reconsider calling today's GCMs "fully physics-based" (86).

Lastly: what precisely does it mean to hold GMT "well below" (14) some temperature threshold? How would we know if we had missed this target? Can this be phased with sufficient precision to allow, say, an insurance contract or legal wager to hinge on

its occurrence? Issues include the duration for which the threshold is exceeded (An instant? A month? A year? A decade?) and how to deal with the imprecision in measuring the global mean temperature, even today. In practice, simply setting the target as an absolute value of GMT, inspired by the agreed 1.5 change, would prove more straightforward both scientifically and legally, even if not politically or diplomatically.

Flato, G., J. Marotzke, B. Abiodun, P. Braconnot, S.C. Chou, W. Collins, P. Cox, F. Driouech, S. Emori, V. Eyring, C. Forest, P. Gleckler, E. Guilyardi, C. Jakob, V. Kattsov, C. Reason and M. Rummukainen, 2013: Evaluation of Climate Models. In: Climate Change 2013: The Physical Science Basis. Contribution of Working Group I to the Fifth Assessment Report of the Intergovernmental Panel on Climate Change [Stocker, T.F., D. Qin, G.-K. Plattner, M. Tignor, S.K. Allen, J. Boschung, A. Nauels, Y. Xia, V. Bex and P.M. Midgley (eds.)]. Cambridge University Press, Cambridge, United Kingdom and New York, NY, USA.

Smith, L.A. (2001) 'Disentangling uncertainty and error: on the predictability of nonlinear systems', in Mees, A.I. (ed.) Nonlinear Dynamics and Statistics, Boston: Birkhauser, 31-64.

Smith, L.A. (2002) 'What might we learn from climate forecasts?', Proc. National Acad. Sci. USA, 4 (99): 2487-2492.

---

## Editor Comment (EC1) · S. Bronnimann (Editor) · 4 Sep 2017

Dear authors,

The open discussion phase of your manuscript is now closed. As you have seen, your paper received some critical comments, which I ask you to consider. Please respond to these comments and then submit a revised version of the manuscript. As to the comment of Reviewer #1, asking whether your paper fits the scope of CP, I would like to assure you that it does fit the scope.

I am looking forward to reading your replies.

Best regards, Stefan Brönnimann Editor

---

## Author Comment (AC1) · 15 Sep 2017

**Authors response to two reviewers and comments of L.A. Smith**

In the following we will respond to the major points given by two reviewers and L.A. Smith. We first give their comments in italic, followed by our response.

**Answers to major points of reviewer # 1 (Peter Thorne)**

*I find the hook to pre-industrial tenuous given that the authors make no attempt to estimate a true pre-industrial based value. They would be better, in my view, to state that they are making an estimate relative to the late 19th Century / early global instrumental record. This would be a fairer reflection of what is actually done and consistent with e.g. IPCC AR5 which deliberately avoided in the published version implying that 1850-1900 constituted pre-industrial as noted in Hawkins et al. Indeed, the final plenary of the WG1 involved a long discussion that I was personally involved in around the topic whereby the parties agreed that pre-industrial was earlier than 1850. It would be unwise, in my view, for the authors to reopen this issue. I note in a couple of places that there are phrases which could imply IPCC used 1850-1900 as pre-industrial, and they did not. Such implications absolutely must be avoided outright in any resubmission.*

*The discussion linking their work to the pre-industrial era would be far better being given exclusively in the Discussion section and, to my view, the authors should remove allusions to providing an estimate relative to pre-industrial earlier than this. Bottom line: They either should estimate relative to true-pre-industrial or be honest with respect to what they are estimating relative to for the paper to be acceptable. As I see it there is no rigorous attempt to estimate changes since pre-industrial. Rather, there is a rigorous attempt to estimate it since 1880 which in itself is useful and valuable. The authors should be honest in this regard and not oversell their work by claiming it's an estimate relative to pre-industrial when it demonstrably is not.*

We agree with the reviewer and will adapt the text in the way he suggests. Our uncertainty and sensitivity analysis is clearly relative to 1880, and not 1850, or relative to the period 1720-1800 (as in Hawkins et al. 2017), or even relative to the period 1400-1800 (as in Schurer et al., July 24, 2017 - Nature Climate Change). The reason we choose for 1880, is (i) data availability and (ii) the increasing uncertainties in GMT estimates for years earlier than 1880. For example, the Hadley Centre estimates the GMT value plus uncertainty in 1900 to be -0.20 [-0.34, -0.06] ℃ (95% confidence limits). For 1850 the GMT estimate is -0.37 [-0.59, -0.16] ℃.

We propose to follow the comment to treat the role of 'pre-industrial' solely in the discussion. We propose to add the results of Schurer et al. (2017) who analyze the role of GHGs, solar radiation and volcanic dust from 1401 onwards. They find that GHGs had a significant effect on global warming if the period 1401-1800 is compared to 1850-1900: from 0.02 to 0.20 ℃ (5-95% confidence limits). If all forcings are combined (GHG, solar, volcanic) they find 0.09 [0.03 - 0.19] °C.

*I also find the Section at the end of the paper alluding to RCPs and end of Century to be out of scope and a distraction. It should either form an integral part of the paper integrated throughout or be dropped. Given journal scope I would lean heavily toward its removal. The year 2100 is not in the past (at least yet)!*

Agreed. We propose to remove 'the future' in Section 5.2, including Figure 5.

*Finally, given the authors apparent desire to explore uncertainty I find the omission of the*
*JMA observational analysis and the NOAA 20CR product odd. I could see a case for omission*
*of 20CR, but I see no logical case why the JMA analysis should be omitted here as it has the*
*same non-peer-reviewed basis as e.g. the Berkeley global (not land, but global) estimate.*
*JMA uses a fundamentally distinct set of SSTs and so would better span uncertainty that the*
*authors lament in Section 2.1.*

We have studied the global series of the Japan Meteorological Agency (JMA). There is a
simple practical problem, however: this series starts in 1891. Thus, we miss the important
period 1880-1890. Next to that, there are only texts in Japanese which explain how this data
product is composed. Finally, studies which show the JMA series are limited. Our choice for
five GMT data products as described in Section 2.1 is consistent with recent studies such as
Medhaug et al. (2017 - their figure 1a) or Rahmstorf et al. (2017 - their figures 1 and 2).
Therefore, we propose not to add trend analyses for JMA to Table 1.

As a test we estimated linear trends for all five data products shown in Table 1 and
additionally the JMA series. It appears that the JMA incremental value for the period
1891-2016 equals the low end of the five data products we apply in our Table 1 (i.e., the
incremental value of the HadCRUT4 series). Thus, the incremental value based on the JMA
series, does not fall outside the range of values based on HadCRUT4, NASA, NOAA,
HadCRUT4 adapted by Cowtan and Way, and BEST.

As for the NOAA 20CR series we have two arguments for not adding it to our study. First, the
20CR series covers the period 1851-2011. Thus, data for the important period 2012-2016 are
missing. Second, the series is a combination of modeling (weather prediction models) and
data. For our study we prefer to make a distinction between GMT series directly derived from
temperature registrations and models, be it GCMs or weather prediction models.

We propose to name these arguments in the first item of the discussion section.

*Minor comment #12. Line 202 you should clarify what the implications of ignoring this are or,*
*preferably, perform the extra work necessary for its inclusion. Presumably the impact would*
*be artificially reduced uncertainty ranges? In which case is it really safe to ignore this*
*issue? I'm not entirely convinced and would suggest that extra work leading to its*
*inclusion is instead warranted. Even if it ends up showing no change it would make*
*the piece more robust. As you yourselves state the effect is statistically significant, in*
*which case it really should be included.*

Agreed. We propose to show the effect on correcting for this small but significant AR(1)
correlation in the innovation series of our Kalman filter model.

**Answers to major points of Anonymous Referee #2**

*This study considers the question of estimating by how much global temperatures have*
*changed since 'pre-industrial' times, assessing the uncertainty in different trend models*
*and due to different global temperature datasets. The analysis is interesting, though*
*the results are not too surprising. However, I have some major concerns:*

*1) Framing: the authors emphasize repeatedly that they are estimating changes since*
*a particular baseline and implying that this is what the Paris agreement meant by*
*'preindustrial'.*

*This is not the case - the introduction of Hawkins et al. (which the authors*
*cite) discusses this issue at length. In addition, Schurer et al. (2017, NCC) was very*
*recently published, highlighting again that there was likely some additional warming*
*due to anthropogenic factors before 1850. The authors may also like to examine Otto*
*et al. (2015) for an alternative approach to estimating the warming since the 19th century.*
*The text in the discussion on this topic is appropriate however.*

Agreed. We propose to treat the topic of 'pre-industrial' more clearly in the discussion section,
as we pointed out in our response to Reviewer #1. We will add the references to Schurer et al.
(2017) and Otto et al. (2015). Consequently, we will address their findings that GHGs had a
significant effect on global warming if the period 1401-1800 is compared to 1850-1900: from
0.02 to 0.20 ºC (5-95% confidence limits). If all forcings are combined (GHG, solar, volcanic)
they find 0.09 [0.03 - 0.19] °C.

*2) Terminology: some of the phrasing is very confusing when referring to and/or*
*distinguishing between natural *forced* variability (volcanic, solar) and internal *unforced**
*variability. These terms are sometimes mixed and it's not always clear what the authors*
*mean. For example, in the abstract (and L86) the authors claim the models are*
*corrected for natural variability, when they mean the forced component, but the introduction*
*uses natural variability to mean both forced and unforced variations. On L133,*
*the authors refer to the 'historicalNat' runs 'for natural unforced variability', which is not*
*true - those runs include both natural forced and internal unforced variations as the*
*next sentence correctly states. Variability is also used for the spread or range between*
*different estimates, adding further confusion. The authors should carefully check each*
*use of this type of phrasing and make it far more precise.*

Agreed. We propose to check the phrasing of 'natural variability' carefully, this in
combination with the terms 'forced' or 'unforced' or both, 'internal variability' and 'spread'.

Additionally, we propose to treat the role of natural unforced variability and natural forced
variability (i.e., the role of changes in irradiance of the sun and changes in volcanic activity)
separately in a second item in the discussion section.

The trend analyses as given in our Table 1 are based on the IPCC definition of climate change
(Glossary AR5): anthropogenic forcing combined with decadal to centennial natural
variability. However, UNFCCC defines climate change as originating from GHG forcing
only. In their philosophy we could argue that the Paris limits of 1.5 and 2.0 C should originate
solely from anthropogenic forcing. We propose to quantify this second view on the Paris
limits.

To do so we make use of the recent study of Schurer et al. (2017, their figures S2 and S3), and
the lower panel of figure 4 in our manuscript. Next to that we estimated the role of volcanos
in a time-series setting by extending the Integrated Random Walk (IRW) model. For details
we will refer to Visser and Molenaar (1995) and Visser et al. (2015).

It shows that the incremental values shown in Table 1 for the IRW trend are 0.04 ºC degree
lower. If estimated in combination with the OLS straight line, i.e. a regression model with one
explanatory variable, estimates are 0.02 °C lower than those shown in table 1. This effect,
although small, will be due to the Krakatoa eruption in the period 1880-1890.

The indicator for volcanic dust is taken from NASA: aerosol optical depth (AOD). See graph
below:

[Figure]

*3) GCM analysis: the 106 members used cannot be 'one per model' as there were not*
*that many models in CMIP5. It's not clear what the authors have used here - there must*
*be more than one historical part of the runs for some of the models.*

The reviewer addresses a good point. What we meant here is that we used one member per
model, **given the use of a specific RCP scenario**. Thus, we have used 42 members for
emission scenario RCP4.5, 25 members for emission scenario RCP6.0 and 39 members for
emission scenario 8.5, making up a total of 106 members. We propose to clarify this in the
text.

*There are also 43 piControls on Climate Explorer, and very few are less than 200 years, not*
*only the 20 that the authors have used - why have they not used the others?*

Agreed. We have calculated all AR(1) coefficients for all 41 piControl runs, available in the
KNMI Climate Explorer. Three of those runs showed a jump or a strong linear trend over the
simulation period (varying from 200 to 1000 years). We omitted these. For the remaining 38
runs we have omitted the lowest two AR(1) coefficient estimates (lying around 0.0) and the two highest estimates (lying around 0.75). The remaining range equals the range given in our manuscript: [0.28 - 0.60]. We propose to adapt the text for this finding.

*Also, in section 3.2, the authors could use the AR(1) value from each model's own control run to fit a spline to the historical run of that same model, rather than assume the same across every model. Also, how has the correction for natural forcings been applied (L250)? Has the mean across the historicalNat runs been subtracted from each historical run? If so, this is inconsistent as the response to volcanic eruptions varies significantly across models.*

In our revision we propose to give values for smoothing by splines with φ=0.28 and φ=0.60, similar to shown in our figure 3. Period: 1861-2016. This gives a small change in the upper panel of our figure 4. The spread is for both smoothing options identical ± 0.50 C (2σ). The mean value of all 106 increments is 1.15 for the smoothing option with φ=0.28 and 1.00 for φ=0.60.

**Answers to major points of L.A. Smith**

*Visser et al (2017) provide an interesting and insightful discussion of signal detection in global mean temperature (GMT), focusing on the 1.5 degree target of the Paris Agreement of 2015. This paper could be made more informative by further consideration of three topics: (1) clarifying what is meant by "signal" and by "noise", and more specifically how (whether) natural variability can be "corrected for" in an evolving nonlinear system, (2) implications of using CMIP5 models, given that those models display a wide range of values for today's GMT, and (c) a cleaner definition of how one would detect failure to stay "well below" a temperature target, or to exceed it. These points are expanded upon below.*

*Specific Comments*
*"Natural variability" is said to be a dominant source of uncertainty which has been "corrected for" (24). Although discussions of a climate signal coming "out of the noise" are common, the notions underlying the distinction between signal and noise in the climate context is unclear; it is not the traditional distinction of observational noise superimposed on a imprecisely measured but well-defined signal. Superposition can only be assumed in nonlinear systems given purely observational noise that has no impact on the system: natural variability, internal variability and the like alter the dynamics, and thus the "signal" itself, if such a separation exists (Smith (2001,2002)). A more appropriate conceptualization in nonlinear systems is found in consideration of an ensemble of systems each subject to a common driving and independent realizations of the relevant noise. In this case, the ensemble median would provide a well-defined signal while the distribution about it would capture the effects of noise processes. This view is of limited utility in climate science, where there is only one realization (the Earth): particular realizations need not reflect the (unobservable, non-empirical) "signal"; indeed they can diverge arbitrarily far from it. So in no sense can one expect "the" signal to emerge from the noise, given observations of a single realization. While vague appeals to something somewhat reminiscent of an adiabatic change in thermodynamics may be voiced, clear clarification of the meaning of signal and noise in the climate context would be of value.*

*In short: it would be useful to clarify how "natural variability" and "internal variability" might be isolated in the case of a complicated, nonlinear, evolving planetary system.*

*How are we to make sense of the traditional notions of "signal" and "noise" given that*
*the "noise" is not mere observational noise but actually a component of the system*
*dynamics, and given that in nonlinear systems we cannot appeal to a principle of*
*superposition of solutions (Smith, 2002).*

The modeling of climate data by stochastic climate models have been described in Mudelsee
(2014, sections 2.5.1 and 2.6). He describes the suitability of climate modeling with AR(1)
processes (and the more general ARIMA models as well) to describe the persistence in data.

The reviewer is right that correlated noise is not the same as climate variability arising from
nonlinear systems. However, statistical modeling has proven fruitful in a wide field of
ecological modeling. To stick to the modeling of global mean temperatures, we refer to our
review of (statistical) trend analyses in the peer-reviewed literature in the Supplementary
Material section of our manuscript (table S.1). Furthermore, Visser et al. (2015) show in their
table 1 that researchers in the field of sea level rise apply 30 trend methods for quantifying
"the signal" in sea level data, all with different mathematical formulations.

Note: we do not use trend models for **prediction**. Next to that, projections up to the year 2100
will be removed.

*It is also worth noting that the statistics community and the physical science community*
*often hold very different notions of what a trend is: for the first, it is a statistically consistent*
*combination of two well-defined models (the trend model and the noise model), while for the*
*second it is merely a systematic, often obvious drift. Statisticians require, and quantify,*
*consistency between these two components, and reject identification of a trend if that*
*consistency is lacking. Physical scientists often require the observations to look trendy, and*
*the ability to reject simple statistical models given the data, when those models are known by*
*construction not to admit a trend. The second bar is much lower.*

*The claim that modelling groups "have not been very successful in tuning to the observed*
*trend" (299) suggests some knowledge as to how large the spread would be in*
*the absence of each group knowing the observed trend (aiming for the same target).*
*It has been argued elsewhere that knowledge of such spread would be very useful to*
*have if, perhaps, impractical to obtain.*

*Visser et al (2017) state that "mean progression derived from GCM-based GMTs appear*
*to lie within the range of the trend-dataset combinations" (311). It would be interesting*
*to see the variations among individual CMIP5 simulations (not the mean over*
*them, but their distribution). The IPCC AR5 reports that variations in the global mean*
*temperature of today's CMIP5 GCMs have a range exceeding 2.5 degrees (see right*
*side axis labels of Figure 9-08 of Flato et al (2013)); what are the implications of our*
*best models showing a range of GMT almost twice the 1.5 degree target? Physical*
*and biological processes are driven by actual temperature, not anomalies. Given the*
*current (limited) level of realism in these models, and the fact there is a great deal more*
*in them than their basis in physical understanding, the authors might wish to reconsider*
*calling today's GCMs "fully physics-based" (86).*

The upper panel of figure 4 shows in part what the reviewer asks for. We will discuss the
implication of the wide range of incremental values at the end of the discussion section. Here,
we will argue that GCM simulations are less suited for tracking the signal in GMTs due to their wide range. Another argument will be that GCM simulations in CMIP5 are up to date up
to the year 2005. Estimates for the period 2006-2016 are less reliable.
We propose to add the important comment  of the reviewer that GCMs give a wide range of
estimates for the global temperature over the period 1961-1990. Not as anomalies but in
**absolute temperatures**. Indeed, figure 9.8 of the AR5 WGI  report (2013, page 768) shows a
range from 12.6 ºC to 15.3 ºC, based on 36 models. This range is almost the double of the
1.5 ºC limit..  Also see figure 1 upper panel in Hawkins and Sutton 2016 BAMS 963-980.
Finally, we propose to remove the expression that GCMs are 'fully physics based'. That is,
indeed, not true.
*Lastly: what precisely does it mean to hold GMT "well below" (14) some temperature*
*threshold? How would we know if we had missed this target? Can this be phased*
*with sufficient precision to allow, say, an insurance contract or legal wager to hinge on its*
*occurrence? Issues include the duration for which the threshold is exceeded (An*
*instant? A month? A year? A decade?) and how to deal with the imprecision in measuring*
*the global mean temperature, even today. In practice, simply setting the target*
*as an absolute value of GMT, inspired by the agreed 1.5 change, would prove more*
*straightforward both scientifically and legally, even if not politically or diplomatically.*
Good point. However, we propose to remove Section 5.2 where we extend the historical
analysis to the year 2100. Therefore, this important comment is not directly applicable to our
revised text.
**Additional changes to a revised manuscript**
Due to the comments given by the reviewers we propose to restyle the discussion section  to
improve readability. The following items are addressed (in this order):
- Data products, trend models and GCM simulations. Would the results presented here,
change if (i) other data products would have been chosen (such as JMA), (ii) other
trend models should have been chosen, or (iii) other GCM simulations would have
been chosen (such as blended simulations)?
- Correcting for forced and unforced natural variability. This is a new item which
addresses the question if we should filter a GMT series for short term natural
variability, complying with the IPCC definition of climate change, or that we should
filter GMTs for all natural forcings, complying with the definition of climate change
of UNFCCC.
- Choosing a pre-industrial baseline. Addresses the consequences of results presented by
Hawkins et al. (2017) and Schurer et al. (2017).
- Policy recommendation. This is a the slightly extended text taken from Section 5.2.
Furthermore, we remove section 5.2.

---

## Author Response (AR1)

**Authors response to two reviewers and to the comments of L.A. Smith**

In the following we respond to the comments given by two reviewers and L.A. Smith. We first give their comments in italic, followed by our response. Where relevant we will point to the
lines in the *revised* manuscript where changes have taken place.

**Answers to comments of reviewer # 1 (Peter Thorne)**

*I find the hook to pre-industrial tenuous given that the authors make no attempt to*
*estimate a true pre-industrial based value. They would be better, in my view, to state that they are making an estimate relative to the late 19th Century / early global instrumental record. This would be a fairer reflection of what is actually done and consistent with e.g. IPCC AR5 which deliberately avoided in the published version implying that 1850-1900 constituted pre-industrial as noted in Hawkins et al. Indeed, the final plenary of the WG1*
*involved a long discussion that I was personally involved in around the topic whereby the parties agreed that pre-industrial was earlier than 1850. It would be unwise, in my view, for the authors to reopen this issue. I note in a couple of places that there are phrases which could imply IPCC used 1850-1900 as pre-industrial, and they did not. Such implications absolutely must be avoided outright in any resubmission.*
*The discussion linking their work to the pre-industrial era would be far better being given exclusively in the Discussion section and, to my view, the authors should remove allusions to providing an estimate relative to pre-industrial earlier than this. Bottom line: They either should estimate relative to true-pre-industrial or be honest with respect to what they are*
*estimating relative to for the paper to be acceptable. As I see it there is no rigorous attempt to estimate changes since pre-industrial. Rather, there is a rigorous attempt to estimate it since 1880 which in itself is useful and valuable. The authors should be honest in this regard and not oversell their work by claiming it's an estimate relative to pre-industrial when it demonstrably is not.*
We agree with the reviewer and will adapt the text in the way he suggests. Our uncertainty and sensitivity analysis is relative to 1880, and not 1850, or relative to the period 1720-1800 (as in Hawkins et al. 2017), or even relative to the period 1400-1800 (as in Schurer et al., July 24, 2017 - Nature Climate Change). The reason we choose for 1880, is (i) data availability and (ii)
the increasing uncertainties in GMT estimates for years earlier than 1880. For example, the Hadley Centre estimates the GMT value plus uncertainty in 1900 to be -0.20 [-0.34, -0.06] ºC (95% confidence limits). For 1850 the GMT estimate is -0.37 [-0.59, -0.16] ºC. We also remove any text suggesting that IPCC has defined pre-industrial levels (but simply refer objectively to the pragmatic reference to a fixed period as done by
IPCC). E.g., see lines 295-296.

In the revised text we follow the comment to treat the role of 'pre-industrial' solely in the discussion. See lines 293-313. We added the results of Schurer et al. (2017) who analyze the role of GHGs, solar radiation and volcanic dust from 1401 onwards. They find that GHGs had a significant effect on global warming if the period 1401-1800 is compared to 1850-1900: from 0.02 to 0.20 ºC (5-95% confidence limits). If all forcings are combined (GHG, solar, volcanic) they find 0.09 [0.03 - 0.19] °C.

We explicitly note that the results in our table 1 are relative to 1880, and not 1850, 1720 or 1401. See lines 298-302.

*I also find the Section at the end of the paper alluding to RCPs and end of Century to be out of scope and a distraction. It should either form an integral part of the paper integrated throughout or be dropped. Given journal scope I would lean heavily toward its removal. The year 2100 is not in the past (at least yet)!*

Agreed. We have removed 'the future' in Section 5.2, including Figure 5.

*Finally, given the authors apparent desire to explore uncertainty I find the omission of the JMA observational analysis and the NOAA 20CR product odd. I could see a case for omission of 20CR, but I see no logical case why the JMA analysis should be omitted here as it has the same non-peer-reviewed basis as e.g. the Berkeley global (not land, but global) estimate. JMA uses a fundamentally distinct set of SSTs and so would better span uncertainty that the authors lament in Section 2.1.*

We have studied the global series of the Japan Meteorological Agency (JMA) carefully. There is a however a practical problem: the JMA series start in 1891. Thus, we miss the important period 1880-1890. In addition, the meta data is only in Japanese. Finally, studies which show the JMA series are limited. E.g., it is not named in IPCC (2013, Ch. 2). We therefore have chosen for five GMT data products as described in Section 2.1 and we did not add trend analyses for JMA to Table 1. This choice is consistent with recent studies such as Medhaug et al. (2017 - their figure 1a) or Rahmstorf et al. (2017 - their figures 1 and 2). These studies use the 5 datasets as we do. Nevertheless, the referee addresses a relevant issue.

As a test we estimated linear trends for all five data products shown in Table 1 and additionally the JMA series. It appears that the JMA incremental value for the period 1891-2016 equals the low end of the five data products we apply in our Table 1 (i.e., the incremental value of the HadCRUT4 series). Thus, the incremental value based on the JMA series, does not fall outside the range of values based on HadCRUT4, NASA, NOAA, HadCRUT4 adapted by Cowtan and Way, and BEST.

We have hesitated to include this argumentation and analysis in the manuscript. However, the revised article is already loaded with results and sensitivity analyses. It would influence the readability of the text in a negative way, we feel.

As for the NOAA 20CR series we have two arguments for not adding it to our study. First, the 20CR series covers the period 1851-2011. Thus, data for the important period 2012-2016 are missing. Second, the series is a combination of modeling (weather prediction models) and data. For our study we prefer to make a distinction between GMT series directly derived from temperature registrations and models, be it GCMs or weather prediction models.

We note that we give a number of details on data products in the new lines 117-133. These details are certainly not exhaustive. However, how data products are made, including complex interpolation schemes, etc etc, is not the topic of this article. To compensate for this, we cite all relevant literature.

*I have a number of further comments, suggestions and requests which I refer to in the order they arise chronologically below:*

*1. Line 19 per above remove 'and what is 'pre-industrial'?' as you make no attempt to robustly address that question.*

Removed

*2. Line 47 GMTs have .... So following the 21st*

Okay, changed.

*3. Lines 53-56 do not reflect the IPCC approach. This was not an attempt to inform on*
*post-pre-industrial changes and it did not involve expert judgement. Rather the*
*stated range is the range of available estimates and their uncertainties after*
*correcting for AR(1) and using OLS. The text here significantly overcomplicates both*
*what was done and why. As the IPCC author who undertook the lead on this analysis*
*I can assure the authors it was not as complicated as they imply here. This should be*
*revised to reflect the actual process.*

We changed the text in a way that IPCC did not pretend to give GMT trend progression since pre-industrial. See new lines 53-58. However, we decided to keep the words 'and expert judgement'. This might surprise both reviewers #1 and # 2. Two of the authors of this article were also at the IPCC discussions on this point (Bram Bregman and Arthur Petersen). The addition of 'expert judgment' was proposed by Petersen and agreed upon. The role of judgments here is shown in Box 2.2 of IPCC (2013 - pages 179 and 180).

*4. If retained (and note earlier major suggestion to move this to discussion) line 55 forwards should constitute the beginning of the paragraph currently starting line 57.*

Agreed and changed

*5. Line 73 or similar do not*

Agreed and changed

*6. Line 73. Reader will ask so what? You need to be explicit that the approach limitations matter in a period of rapid change.*

We now explain this point in lines 75-77.

*7. Line 76 progression to specific (remove allusion to pre-industrial per major comment)*

Removed

*8. Line 88 the main one being*

Done.

*9. Lines 108 to 121 omit the by far largest overlap of all in that the NOAA and NASA products are based on identical underlying land and ocean datasets differing solely in the applied post-processing. This needs to be acknowledged for this discussion to be acceptable. More generally this discussion is incomplete. It needs to be expanded and may be better if supported by a table.*

The referee addresses an important point, but, as explained above the exact construction of data products is not the topic of this article. We apply the data sets in the same way as for GCM

simulation data sets. To describe the details of all applied datasets (from observations and model results) would be a huge complicated effort and beyond the scope of this work. For example, the differences between NOAA and NASA are quite complicated given the corrections of Karl et al. (2015). However, to address this useful point of the referee, we have carefully checked the completeness of the references and emphasized our approach in the text.

*10. Feels odd not to discuss and cite Cowtan et al at lines 125-127*

Agreed. We added the reference, see line 140.

*11. Lines 155-157. First please clarify whether the AR1 factors are calculated on the annual series. This is important information that is being omitted. Secondly, even at annual scales the AR(1) is primarily an artefact of variability and not forcing so the assertion here is wrong as you note in lines 175-177. Your two cheek-to-jowl statements here cannot both be right. The AR arising from variability is the correct*
*one here. Year-to-year autocorrelation does not arise mainly due to forcing.*

Agreed, we added 'annual series' in line 171. Furthermore, we removed 'forcing' and replaced by persistence in natural processes: lines 172-173.

*12. Line 202 you should clarify what the implications of ignoring this are or, preferably, perform the extra work necessary for its inclusion. Presumably the impact would be artificially reduced uncertainty ranges? In which case is it really safe to ignore this issue? I'm not entirely convinced and would suggest that extra work leading to its inclusion is instead warranted. Even if it ends up showing no change it would make*
*the piece more robust. As you yourselves state the effect is statistically significant, in which case it really should be included.*

Agreed. We now show the effect on correcting for this small but significant AR(1) correlation in the innovation series of our Kalman filter model. See lines 223-225. Also uncertainty bands
in table 1 are adapted accordingly.

*13. Lines 224-227. This is true a. for this particular period and b. this particular small (and non-independent as noted in Section 2.1) draw from the broad range of plausible means by which to estimate historical changes in GMT. Hence I believe this*
*statement oversimplifies the issues and as a result is more confident than is, in reality, warranted. The findings do not have the universality implied here and may not even be true if we instead had a further draw from the sample of plausible*

*approaches to estimating GMTs from observations. Here, JMA's inclusion may fundamentally alter this finding which would imply non-robustness*

We feel that the revised version contains a number of lines which show that these results are based on this choice of GMT products, this choice of trend methods and this choice of 1880 (in stead of 1850, 1720 or 1401). We discussed the point of *not* adding the JMA series above.

**Answers to comments of Anonymous Referee #2**

*This study considers the question of estimating by how much global temperatures have changed since 'pre-industrial' times, assessing the uncertainty in different trend models and due to different global temperature datasets. The analysis is interesting, though the results are not too surprising. However, I have some major concerns:*

*1) Framing: the authors emphasize repeatedly that they are estimating changes since a particular baseline and implying that this is what the Paris agreement meant by 'preindustrial'.*

*This is not the case - the introduction of Hawkins et al. (which the authors cite) discusses this issue at length. In addition, Schurer et al. (2017, NCC) was very recently published, highlighting again that there was likely some additional warming due to anthropogenic factors before 1850. The authors may also like to examine Otto et al. (2015) for an alternative approach to estimating the warming since the 19th century. The*

*text in the discussion on this topic is appropriate however.*

Agreed. We now treat the topic of 'pre-industrial' more clearly in the discussion section, as we pointed out in our response to Reviewer #1. We added the references to Schurer et al. (2017) and Otto et al. (2015). Consequently, we address their findings that GHGs had a significant effect on global warming if the period 1401-1800 is compared to 1850-1900: from 0.02 to 0.20
ºC (5-95% confidence limits). If all forcings are combined (GHG, solar, volcanic) they find 0.09 [0.03 - 0.19] °C. See lines 293-313. Otto et al. is named in line 83.

*2) Terminology: some of the phrasing is very confusing when referring to and/or distinguishing between natural \*forced\* variability (volcanic, solar) and internal \*unforced\**
*variability. These terms are sometimes mixed and it's not always clear what the authors mean. For example, in the abstract (and L86) the authors claim the models are corrected for natural variability, when they mean the forced component, but the introduction uses natural variability to mean both forced and unforced variations. On L133,*

*the authors refer to the 'historicalNat' runs 'for natural unforced variability', which is not*

*true - those runs include both natural forced and internal unforced variations as the*
*next sentence correctly states. Variability is also used for the spread or range between*
*different estimates, adding further confusion. The authors should carefully check each*
*use of this type of phrasing and make it far more precise.*

Agreed. We checked the phrasing of 'natural variability' carefully, this in combination with the terms 'forced' or 'unforced' or both, 'internal variability' and 'spread'. See also the reply to reviewer #1 who commented on two sentences with three times the word 'variability'.

Additionally, we now treat the role of natural unforced variability and natural forced variability
(i.e., the role of changes in irradiance of the sun and changes in volcanic activity) separately in a third item in the discussion section: lines 315-336.

The trend analyses as given in our Table 1 are based on the IPCC definition of climate change (Glossary AR5): anthropogenic forcing combined with decadal to centennial natural variability.
However, UNFCCC defines climate change as originating from GHG forcing only. In their philosophy we could argue that the Paris limits of 1.5 and 2.0 C should originate solely from anthropogenic forcing. We now quantify this second view on the Paris limits.

To do so we make use of the recent study of Schurer et al. (2017, their figures S2 and S3), and
the lower panel of figure 4 in our original manuscript. Next to that we estimated the role of volcanos in a time-series setting by extending the Integrated Random Walk (IRW) model. For details we refer to Visser and Molenaar (1995) and Visser et al. (2015).

See lines 315-322 and the **new** table SM.2, in the Supplementary Material section.
It shows that the incremental values shown in Table 1 for the IRW trend are 0.04 ºC degree lower. If estimated in combination with the OLS straight line, i.e. a regression model with one explanatory variable, estimates are 0.02 °C lower than those shown in table 1. This effect, although small, will be due to the Krakatoa eruption in the period 1880-1890.
The indicator for volcanic dust is taken from NASA: aerosol optical depth (AOD). See graph below:

[Figure]

This graph is the new figure 5 in the revised text.

*3) GCM analysis: the 106 members used cannot be 'one per model' as there were not*
*that many models in CMIP5. It's not clear what the authors have used here - there must*
*be more than one historical part of the runs for some of the models.*

The reviewer addresses a good point. What we meant here is that we used one member per
model, **given the use of a specific RCP scenario**. Thus, we have used 42 members for emission
scenario RCP4.5, 25 members for emission scenario RCP6.0 and 39 members for emission
scenario 8.5, making up a total of 106 members. We clarify this in the text: lines 143-146.
*There are also 43 piControls on Climate Explorer, and very few are less than 200 years, not*
*only the 20 that the authors have used - why have they not used the others?*

Agreed. We have calculated all AR(1) coefficients for all 41 piControl runs, available in the
KNMI Climate Explorer (note: there are 41, not 43). Three of those runs showed a jump or a
strong linear trend over the simulation period (varying from 200 to 1000 years). We omitted
these. For the remaining 38 runs we have omitted the lowest two AR(1) coefficient estimates
(lying around 0.0) and the two highest estimates (lying around 0.75). The remaining range equals the range given in our manuscript: [0.28 - 0.60]. We have adapted the text for this finding. See lines 192-194.

*Also, in section 3.2, the authors could use the AR(1) value from each model's own control run to fit a spline to the historical run of that same model, rather than assume the same across every model.*

In our revision we give values for smoothing by splines with φ=0.28 and φ=0.60, similar to shown in our figure 3. Period: 1861-2016. This gives a small change in the upper panel of our old figure 4. The spread is for both smoothing options identical ± 0.50 ℃ (2σ). The mean value of all 106 increments is 1.15 for the smoothing option with φ=0.28 and 1.00 for φ=0.60. See lines 261-266.

*Also, how has the correction for natural forcings been applied (L250)? Has the mean across the historicalNat runs been subtracted from each historical run? If so, this is inconsistent as the response to volcanic eruptions varies significantly across models.*

In the revised text we do not correct GCM simulations anymore. The reason is that inferences in sections 3.1 and 3.2 would become inconsistent: estimates in §3.1 are not corrected for solar and volcanic forcings either.

***Smaller points:***

*L47 - 21th -> 21st*

Done.

*L56 - this uncertainty does not include expert judgement*

We discussed this point in our answer to reviewer #1.

*L87 - this sentence could be read to imply that GCMs should have 'priority' over the observations for answering the question of how much the surface has warmed, but I don't think the authors mean that?*

We removed this wording.

*L146 - I'm not sure the UNFCCC would suggest that pre-industrial should be defined when a particular dataset happens to begin?*

In the revised version we name UNFCCC in the context of warming definitions, in contrast to the definition of IPCC: lines 78-84. What we meant in the line L146 is that 'pre-industrial' is often denoted as a *period*. However, in the context of trend modeling one does not define any period. The outcome of the analysis solely depends on the sample period chosen, thus 1880-2016, 1850-2016 or 1401-2016, or similar. We did not change the text here.

*L163-4 - do you need all five of those references to the lead author's previous papers?*

We removed two references here. See line 178.

*L191 - this not appropriate for a scientific analysis*

Okay, removed.

*L216 - delete 'is'?*

Done.

*L220 & L307 - three 'variability' in the same sentence, all referring to slightly different things?! The first is a 'range'? The sentences following this are also not clear.*

Agreed. We have replaced 'variability' by 'range' on both places. See lines 243-246.

*L279 - not sure this is quite true - there is a signal of these eruptions in the observations but it is probably weaker than the GCMs suggest. But I agree this is probably partly a*
*coverage issue.*

We did not change the text here.

*L288 - most of the time, the models are not tuned to the trend, but are tuned on the*
*mean present-day climate state.*

Agreed.

*L303 - is 'best guess' the appropriate term?*

We replace 'best guess' to 'trend' .

*L315 - it would be useful to use other historical ensemble members to check this statement (where they exist), as they provide another estimate of the change in temperature*
*for the same model.*

Agreed, but we confined our analysis to these 106 simulations from CMIP5 (and discuss the role of 'tas' versus 'blended'). Indeed, new simulations will give new insights, that is how science works. Our main point for choosing data products and not GCM output is given in the new lines
348-354. Hopefully, the wide spread in incremental values as shown in our new figure 4 will become less wide. But we cannot know this at this moment.

**Answers to major points of L.A. Smith**

*Visser et al (2017) provide an interesting and insightful discussion of signal detection in*
*global mean temperature (GMT), focusing on the 1.5 degree target of the Paris Agreement of 2015. This paper could be made more informative by further consideration of three topics: (1) clarifying what is meant by "signal" and by "noise", and more specifically how (whether) natural variability can be "corrected for" in an evolving nonlinear system, (2) implications of using CMIP5 models, given that those models display a wide range of values for today's GMT,*
*and (c) a cleaner definition of how one would detect failure to stay "well below" a temperature target, or to exceed it. These points are expanded upon below.*

*Specific Comments*
*"Natural variability" is said to be a dominant source of uncertainty which has been "corrected*
*for" (24). Although discussions of a climate signal coming "out of the noise" are common, the notions underlying the distinction between signal and noise in the climate context is unclear; it is not the traditional distinction of observational noise superimposed on a imprecisely measured but well-defined signal. Superposition can only be assumed in nonlinear systems given purely observational noise that has no impact on the system: natural*
*variability, internal variability and the like alter the dynamics, and thus the "signal" itself, if such a separation exists (Smith (2001,2002)). A more appropriate conceptualization in nonlinear systems is found in consideration of an ensemble of systems each subject to a common driving and independent realizations of the relevant noise. In this case, the ensemble median would provide a well-defined signal while the distribution about it would capture the effects of*
*noise processes. This view is of limited utility in climate science, where there is only one realization (the Earth): particular realizations need not reflect the (unobservable, non-*

*empirical) "signal"; indeed they can diverge arbitrarily far from it. So in no sense can one expect "the" signal to emerge from the noise, given observations of a single realization. While vague appeals to something somewhat reminiscent of an adiabatic change in thermodynamics*
*may be voiced, clear clarification of the meaning of signal and noise in the climate context would be of value.*

*In short: it would be useful to clarify how "natural variability" and "internal variability" might be isolated in the case of a complicated, nonlinear, evolving planetary system.*
*How are we to make sense of the traditional notions of "signal" and "noise" given that the "noise" is not mere observational noise but actually a component of the system dynamics, and given that in nonlinear systems we cannot appeal to a principle of superposition of solutions (Smith, 2002).*

The modeling of climate data by stochastic climate models have been described in Mudelsee (2014, sections 2.5.1 and 2.6). He describes the suitability of climate modeling with AR(1) processes (and the more general ARIMA models as well) to describe the persistence in data.

The reviewer is right that correlated noise is not the same as climate variability arising from
nonlinear systems. However, statistical modeling has proven fruitful in a wide field of ecological modeling. To stick to the modeling of global mean temperatures, we refer to our review of (statistical) trend analyses in the peer-reviewed literature in the Supplementary Material section of our manuscript (table S.1). Furthermore, Visser et al. (2015) show in their table 1 that researchers in the field of sea level rise apply 30 trend methods for quantifying "the
signal" in sea level data, all with different mathematical formulations.

Note: we do not use trend models for **prediction**. Next to that, projections up to the year 2100 are removed in the revised text.

*It is also worth noting that the statistics community and the physical science community often hold very different notions of what a trend is: for the first, it is a statistically consistent combination of two well-defined models (the trend model and the noise model), while for the second it is merely a systematic, often obvious drift. Statisticians require, and quantify, consistency between these two components, and reject identification of a trend if that*
*consistency is lacking. Physical scientists often require the observations to look trendy, and the ability to reject simple statistical models given the data, when those models are known by construction not to admit a trend. The second bar is much lower.*

*The claim that modelling groups "have not been very successful in tuning to the observed*
*trend" (299) suggests some knowledge as to how large the spread would be in*

*the absence of each group knowing the observed trend (aiming for the same target).*
*It has been argued elsewhere that knowledge of such spread would be very useful to*
*have if, perhaps, impractical to obtain.*

*Visser et al (2017) state that "mean progression derived from GCM-based GMTs appear*
*to lie within the range of the trend-dataset combinations" (311). It would be interesting*
*to see the variations among individual CMIP5 simulations (not the mean over*
*them, but their distribution). The IPCC AR5 reports that variations in the global mean*
*temperature of today's CMIP5 GCMs have a range exceeding 2.5 degrees (see right*
*side axis labels of Figure 9-08 of Flato et al (2013)); what are the implications of our*
*best models showing a range of GMT almost twice the 1.5 degree target? Physical*
*and biological processes are driven by actual temperature, not anomalies. Given the*
*current (limited) level of realism in these models, and the fact there is a great deal more*
*in them than their basis in physical understanding, the authors might wish to reconsider*
*calling today's GCMs "fully physics-based" (86).*

The upper panel of the old figure 4 shows in part what the reviewer asks for. We discuss the
implication of the wide range of incremental values in the new discussion section 4.2. Here, we
argue that GCM simulations are less suited for tracking the signal in GMTs due to their wide
range. Another argument will be that GCM simulations in CMIP5 are up to date up to the year
2005. Estimates for the period 2006-2016 are less reliable.

We added the important comment  of the reviewer that GCMs give a wide range of estimates
for the global temperature over the period 1961-1990. Not as anomalies but in **absolute**
**temperatures**. Indeed, figure 9.8 of the AR5 WGI  report (2013, page 768) shows a range from
12.6 ºC to 15.3 ºC, based on 36 models. This range is almost the double of the
1.5 ºC limit..  Also see figure 1 upper panel in Hawkins and Sutton 2016 BAMS 963-980.
See lines 351-352.

Finally, we removed the expression that GCMs are 'fully physics based'. That is, indeed, not
true.

*Lastly: what precisely does it mean to hold GMT "well below" (14) some temperature*
*threshold? How would we know if we had missed this target? Can this be phased*
*with sufficient precision to allow, say, an insurance contract or legal wager to hinge onits*
*occurrence? Issues include the duration for which the threshold is exceeded (An*
*instant? A month? A year? A decade?) and how to deal with the imprecision in measuring*
*the global mean temperature, even today. In practice, simply setting the target*
*as an absolute value of GMT, inspired by the agreed 1.5 change, would prove more*

*straightforward both scientifically and legally, even if not politically or diplomatically.*

Good point. However, we propose to remove Section 5.2 where we extend the historical analysis to the year 2100. Therefore, this important comment is not directly applicable anymore to our revised text.

**Additional changes to the revised manuscript**

A new aspect in the revised text is the role of warming definitions. We did not address this point
in the original manuscript. This aspect is now addressed throughout the revised text (the Abstract, introduction, ...).

Furthermore, we have removed section 5.2. All text on future projections has been removed, including figure 5.

**Changes made to the Supplementary Material**

There are three changes made to the Supplementary Material. First, we have added table SM.2.
Here, the role of adding volcanic activity as a regression variable is shown. Second, we have added figure SM.2, which we moved from the main text of the original manuscript. Finally, we have added two regression variables in equation (1) to show how the IRW model is extended from 'trends only' to 'trends plus the influence of explanatory variables'.

[revised manuscript text omitted]

---

## Referee Report (RR1)

**Second round review of *Signal detection in global mean temperatures after "Paris": an uncertainty and sensitivity analysis* by Visser et al.**

Peter Thorne

The authors have undertaken a substantive redraft. The redraft has served to address many of the points raised by three substantive reviews. In particular, the removal of the future looking section and redrafting around defining pre-industrial are helpful. Redrafting has raised further queries which preclude a recommendation of acceptance without further revisions. I outline major and minor points to be addressed below.

**Major points**

1. It feels to me like the paper is not overly long and therefore I would query the value of SI over its incorporation into the main text which would presumably make it easier for the reader to understand the piece as a whole.

2. Related to the prior point there is an uneven degree of specificity given to the descriptions of the different statistical methods employed. This extends both across the main text and the SI. Why is the equation describing OLS methodology given in main text but the other (more complicated!) methods only described qualitatively? I'm not sure that I or a reader could repeat the analysis given the vague descriptions available. Section 2.1 should formally lay out mathematically the approaches employed in a consistent manner. Alternatively, Section 2.1 should lay out each method qualitatively and point to the SI (if retained) where each method should be laid out mathematically. This is necessary for reproducibility of the analysis and results.

3. While sympathetic to the response that you are not trying to exhaustively describe the GMST datasets used I still find the current description inadequate for a reader to properly understand the (in)dependence issues and hence properly interpret your findings. I would suggest that you could add a table that clearly delineates key facets of each dataset that you could reference in place of current text. Such a table may have columnar headings:
   a. dataset name and version
   b. Land product used
   c. SST product used
   d. Interpolation method
   e. Period of record
   f. Key references
   g. Website sourced

   At a minimum, please specify the versions of the GMST products you have used in the text to enable replication. But, I think a table as suggested above would be far more helpful to the reader.

4. The editor is free to over-rule this but I, he and others authored Box 2.2 which you are using to support an expert judgement contention at line 56. I have carefully re-read Box 2.2 and see no reasonable interpretation of the text therein that can support its use to contend expert judgement was involved in the derivation of either the trend estimate or its uncertainty. Please therefore remove this contention. Box

2.2 uncertainty range quoted pertains to choice of dataset and natural variability only. There is no role that expert judgement has in informing that range which formally must arise solely from the choices of dataset and natural variability. These were the only two factors considered in the calculation of the range being quoted and to suggest otherwise and that somehow expert judgement was factored into these numbers when it was not is a substantial misrepresentation of the process involved.

There is undoubtedly uncertainty in how the trend should be calculated as alluded to in Box 2.2 and expanded upon in the Chapter 2 supplement, but: i) this is not expert judgement; ii) this would inflate the range; iii) this is what your paper is getting at. Its fine to make all these points. What is not okay is to suggest that a range quoted in IPCC which is inferred solely and exclusively from the range of available products and the trend fitting uncertainty is somehow in addition fudged by an expert judgement factor. Sorry if I have labored the point, but it is really important to not imply something factually incorrect as to the IPCC process here which may yield issues down the line.

5. The inclusion of the volcanic activity is currently a half-way house with the bulk of the analysis in the SI but in general poorly referenced from / discussed in the main text. If retained steps should be taken to better integrate the analysis more comprehensively into the text.

6. The corrections outlined in lines 208 and 223-225 are introduced without sufficient justification for a reader. I'm unclear what these are myself. If you are applying corrections here for the dof issue then this should be incorporated into your methods section (see earlier comment ref. methods and reproducibility). If not then you need to be far more explicit what these are for and why you are justified in making them. Presently the basis is at best ad hoc to a reader.

**Minor points**
1. Not to belabor the point but the most common acronym in use across the literature for Global Mean Surface Temperatures is GMST and not GMT.
2. Lines 16-17 conclusive as regards methods
3. Line 23 leading observational GM(S)T products
4. Line 25 it is unclear what you mean by both sources of uncertainty are dominated by natural variability
5. Line 47 following the 21$^{st}$
6. Three questions lines 89-95 are actually several questions in many cases and need some refinement to be much clearer to the reader. Line 89 to rather than as for? Third bullet could be simplified rather than 3 Qs.
7. Line 104 exhibit not show
8. Line 131-133 are a non-sequitor. If you mean to include the volcanic loadings in your analysis they need to be better integrated and I would question whether in terms of overall narrative this is the best place to bring this text in.
9. Lines 143-144 should note that over the ten or so years that these are RCP scenario driven the RCPs themselves do not diverge substantively from one another. Rather, RCP scenario divergence grows later in the Century. The authors may consider whether it would be worth picking just RCP4.5 here for this reason which may serve to simplify the analysis?

10. Line 168 (and same equation in SI) it is entirely unclear why the second equality applies. Why is there 125 in that term and why is it squared? If it is the time delta then it should be 137 and not 125.
11. I would have thought that LOTI being both low and high estimate in lines 244-246 should be remarked upon. It is to me an unexpected result. I would not expect the same series to arise both the lowest and highest estimate for this term even across methods. Some further analysis and ensuing discussion of this result would be of interest to the reader.
12. Line 259-260 is unclear what is intended. Please expand for clarity.
13. Line 313 assertion and associated precision is not justified by the prior text. Instead consider "This underestimation is uncertain but could amount to up to 0.1C". This would be consistent with the preceding text.
14. Lines 323-325 I do not follow the logic of the argument given here. Given that solar forcing is dominated by a cyclical component with a small linear aspect I doubt it would greatly confound any of the chosen techniques. If it does then you should show it to justify the decision or at least refer to one or more prescient references to justify the decision.
15. Lines 352-354. I don't follow the logic here. The RCP scenarios are "reasonable". Also, I don't see an argument for accurate simulations but rather accurate forcings up to 2005. The distinction matters.
16. Line 355 Table (typo)
17. Line 394 in addition to instead of rather than
18. Line 399-401 Inconsistent with you earlier assertion of using the Hawkins et al. number which is smaller.
19. Figure 2 requires further explanation in the panel in particular with regard to the middle panels
20. SI line 6 considerably not considerately
21. SI line 15 methodologically
22. SI lines 37-45 needs to make a clear association between the numbers and associated table for this passage to make any sense to the reader.
23. SI line 39 resp?
24. SI line 46 delete second in
25. Table SM2 not referenced anywhere as far as I can tell
26. Figure SM2 has only one panel in the submitted version.

---

## Author Response (AR2)

**Second round review of**

***Signal detection in global mean temperatures after "Paris":***
                                                    ***an uncertainty and sensitivity analysis***   **by Visser et al.**

**1 Comments of Peter Thorne, reviewer #1**

*The authors have undertaken a substantive redraft. The redraft has served to address many of the points raised by three substantive reviews. In particular, the removal of the future looking section and redrafting around defining pre-industrial are helpful. Redrafting has raised further queries which preclude a recommendation of acceptance without further revisions. I outline major and minor points to be addressed below.*

*Major points*
*1. It feels to me like the paper is not overly long and therefore I would query the value of SI over its incorporation into the main text which would presumably make it easier for the reader to understand the piece as a whole.*

We agree with this comment and have incorporated the SI into the main text. We found the review on trend methods too long and placed it in the new Appendix A, directly at the end of the paper.

*2. Related to the prior point there is an uneven degree of specificity given to the descriptions of the different statistical methods employed. This extends both across the main text and the SI. Why is the equation describing OLS methodology given in main text but the other (more complicated!) methods only described qualitatively? I'm not sure that I or a reader could repeat the analysis given the vague descriptions available. Section 2.1 should formally lay out mathematically the approaches employed in a consistent manner. Alternatively, Section 2.1 should lay out each method qualitatively and point to the SI (if retained) where each method should be laid out mathematically. This is necessary for reproducibility of the analysis and results.*

Agreed. We have placed the mathematical description, including formulae in the main text (Section 2.2). See the new equations (1), (2), (3a) and (3b).

*3. While sympathetic to the response that you are not trying to exhaustively describe the GMST datasets used I still find the current description inadequate for a reader to properly understand the (in)dependence issues and hence properly interpret your findings. I would suggest that you could add a table that clearly delineates key facets of each dataset that you could reference in place of current text. Such a table may have columnar headings:*
*a. dataset name and version*
*b. Land product used*
*c. SST product used*
*d. Interpolation method*
*e. Period of record*
*f. Key references*
*g. Website sourced*
*At a minimum, please specify the versions of the GMST products you have used in the text to enable replication. But, I think a table as suggested above would be far more helpful to the reader.*

Agreed. We have added a new table 1 in the format suggested by the reviewer.

*4. The editor is free to over-rule this but I, he and others authored Box 2.2 which you are using to support an expert judgement contention at line 56. I have carefully reread Box 2.2 and see no reasonable interpretation of the text therein that can support its use to contend expert judgement was involved in the derivation of either the trend estimate or its uncertainty. Please therefore remove this contention. Box*

*2.2 uncertainty range quoted pertains to choice of dataset and natural variability only. There is no role that expert judgement has in informing that range which formally must arise solely from the choices of dataset and natural variability. These were the only two factors considered in the calculation of the range being quoted and to suggest otherwise and that somehow expert judgement was factored into these numbers when it was not is a substantial misrepresentation of the process involved.*
*There is undoubtedly uncertainty in how the trend should be calculated as alluded to in Box 2.2 and expanded upon in the Chapter 2 supplement, but: i) this is not expert judgement; ii) this would inflate the range; iii) this is what your paper is getting at. Its fine to make all these points. What is not okay is to suggest that a range quoted in IPCC which is inferred solely and exclusively from the range of available products and the trend fitting uncertainty is somehow in addition fudged by an expert judgement factor. Sorry if I have labored the point, but it is really important to not imply something factually incorrect as to the IPCC process here which may yield issues down the line.*

Okay.. We have removed the text 'and expert judgment'. It is interesting to note that any mathematical (climate) model has a lot of expert judgment in it, made by the designer/programmer. But this is not the topic of this paper.

*5. The inclusion of the volcanic activity is currently a half-way house with the bulk of the analysis in the SI but in general poorly referenced from / discussed in the main text. If retained steps should be taken to better integrate the analysis more comprehensively into the text.*

Agreed. We have incorporated the time-series results using the AOD indicator in the main text. Please see table 3 of the re-revised manuscript and references to eqs. (3a) and (3b).

*6. The corrections outlined in lines 208 and 223-225 are introduced without sufficient justification for a reader. I'm unclear what these are myself. If you are applying corrections here for the dof issue then this should be incorporated into your methods section (see earlier comment ref. methods and reproducibility). If not then you need to be far more explicit what these are for and why you are justified in making them. Presently the basis is at best ad hoc to a reader.*

Okay. We improved the text and added two important reference for this correction method (as applied in IPCC, 2013). Please see lines 173-177 of the new text, and lines 193-194.

**Minor points**

*1. Not to belabor the point but the most common acronym in use across the literature for Global Mean Surface Temperatures is GMST and not GMT.*

Okay. Changed throughout the whole text.

*2. Lines 16-17 conclusive as regards methods*

Done.

*3. Line 23 leading observational GM(S)T products*

Done.

*4. Line 25 it is unclear what you mean by both sources of uncertainty are dominated by natural variability*

Text improved.

*5. Line 47 following the 21st*

Done.

*6. Three questions lines 89-95 are actually several questions in many cases and need some refinement to be much clearer to the reader. Line 89 to rather than as for? Third bullet could be simplified rather than 3 Qs.*

We now made it 4 questions with each question consisting of one sentence. Lines 94-99.

*7. Line 104 exhibit not show*

Done.

*8. Line 131-133 are a non-sequitor. If you mean to include the volcanic loadings in your analysis they need to be better integrated and I would question whether in terms of overall narrative this is the best place to bring this text in.*

Agreed. We moved this text to lines 223-224.

*9. Lines 143-144 should note that over the ten or so years that these are RCP scenario driven the RCPs themselves do not diverge substantively from one another. Rather, RCP scenario divergence grows later in the Century. The authors may consider whether it would be worth picking just RCP4.5 here for this reason which may serve to simplify the analysis?*

Agreed. See also comment of Reviewer #2 as for double counting of simulations. We checked double counting in the 106 simulations. Reviewer #2 is right, there are simulations double. Therefore, we decided to follow the advice here made by Reviewer #1: we choose the 42 simulations with RCP45 with one member per model. This has been changed throughout the text. The new figure 5 now contains 42 increments in stead of 106 increments. The spread and values are almost identical to that made for the 106 simulations! This is not surprising given the double counting. In summary: we have substituted the 106 simulations to 42 simulations and inferences on the results stay almost exactly the same.

*10. Line 168 (and same equation in SI) it is entirely unclear why the second equality applies. Why is there 125 in that term and why is it squared? If it is the time delta then it should be 137 and not 125.*

Correct. This '125' must come from an older copy-past operation ;-[ . The square comes from standard statistics: $var(ax) = a^2 var(x)$ with parameter a any constant. We have added an extra step in equation (1) to make this more clear.

*11. I would have thought that LOTI being both low and high estimate in lines 244-246 should be remarked upon. It is to me an unexpected result. I would not expect the same series to arise both the lowest and highest estimate for this term even across methods. Some further analysis and ensuing discussion of this result would be of interest to the reader.*

A careful look into table 2 shows that three data-trend combinations have the high estimate of ±0.19 C where LOTI is one of them. Therefore, we changed the text a little bit here. See new lines 275-277. Also changed in the conclusion section.

*12. Line 259-260 is unclear what is intended. Please expand for clarity.*

Agreed. We made the text more clear on this point.

*13. Line 313 assertion and associated precision is not justified by the prior text. Instead consider "This underestimation is uncertain but could amount to up to 0.1C". This would be consistent with the preceding text.*

Agreed and done.

*14. Lines 323-325 I do not follow the logic of the argument given here. Given that solar forcing is dominated by a cyclical component with a small linear aspect I doubt it would greatly confound any of the chosen techniques. If it does then you should show it to justify the decision or at least refer to one or more prescient references to justify the decision.*

There is indeed a cyclic component in long-term reconstructions of the TSI series (SORCE data for example). However, the trend that is reflected in GMSTs, is not the cyclic component. See for example Haustein et al. (2017, their figure 1), Schurer et al. (2017, their figure S3) and many other studies. We have added an extra sentence for this point. Please see lines 350-354. Furthermore, we repeated our view on using non-stationary regressors in multiple regression models at the end of the new Appendix A. There is the danger of interpreting correlations as causal relations if we rely solely on statistical models. Any two series with a rising trend correlate high. Therefore, we prefer models with stationary regressors. Also see table A.1.

*15. Lines 352-354. I don't follow the logic here. The RCP scenarios are "reasonable". Also, I don't see an argument for accurate simulations but rather accurate forcings up to 2005. The distinction matters.*

We substituted the word 'simulation' into 'forcing', new line 381.

*16. Line 355 Table (typo)*

Done.

*17. Line 394 in addition to instead of rather than*

Done.

*18. Line 399-401 Inconsistent with you earlier assertion of using the Hawkins et al. number which is smaller.*

We changed the text here slightly, new lines 432-434. Hawkins et al. is on various choices for 'pre-industrial'. Here, we name the data for solar and volcanic forcing as explained in section 4.1.

*19. Figure 2 requires further explanation in the panel in particular with regard to the middle panels*

We find it beyond the scope of this article to explain all details of this graph and particularly the middle panels. We added the reference to Visser (2004) who gives a good and detailed explanation.

*20. SI line 6 considerably not considerately*

Done.

*21. SI line 15 methodologically*

Done.

*22. SI lines 37-45 needs to make a clear association between the numbers and associated table for this passage to make any sense to the reader.*

Agreed. Text highly improved: new lines 811-818.

*23. SI line 39 resp?*

Removed.

*24. SI line 46 delete second in*

Done.

*25. Table SM2 not referenced anywhere as far as I can tell*

This table is now in the main text as table 3. Reference is now.

*26. Figure SM2 has only one panel in the submitted version*

Agreed. Text removed (see caption of new figure 7).

**2 Comments to the anonymous reviewer #2**

*1. Haustein et al. has just been published on a very similar topic and should be discussed:*
*https://www.nature.com/articles/s41598-017-14828-5*

Okay. We have named this new article on a number of places in the new text. New line 74, lines 401-403 and table A.1 as model #27 at the bottom of the table.

*2. line 101 - 'corrected for natural variability' - I don't think this is correct?*

Good comment. Removed.

*3. line 104 - most GCMs are not tuned to the historical period, but to the present climate. Please ensure clarity in this sentence.*

Improved: new line 107.

*4. Are the 106 GCM simulations all different up to 2005? Some groups used the same simulation for the 1861-2005 period and branched off the RCPs in 2005, reducing the number of independent simulations available. Please check and clarify.*

See our comment to Reviewer #1. The reviewer has a good point. There is some double counting in the 106 simulations. We checked that by making a 106 by 106 correlation matrix for the period 1861-2005. Therefore, we followed the hint of Reviewer #1 to use the 42 simulations with RCP4.5 only. See for example the new figure 5. Results differ only as for a few hundreds of a degree.

*5. line 193 - please include the whole range of the AR1 parameter from the piControls, rather than discarding some, as indicated in the Author Response.*

Done, see new lines 206-209.

*6. lines 212-213 - this is not a scientific comment and should not be included.*

Removed.

*7. line 400 - your volcanic contribution is for a warming. Can you clarify why that is?*

We have added in the new line 434 a reference to section 4.1. We suggest an explanation for this in lines 348-349.

*8. Please ensure the version numbers for the observational datasets used are included.*

Version numbers are given in the new table 1.